# A universal SPI candidate distribution function for observations and simulations

Patrick Pieper[1], André Düsterhus[2], and Johanna Baehr[1]

[1]Institute for Oceanography, Center for Earth System Research and Sustainability, Universität Hamburg, Hamburg, Germany
[2]ICARUS, Department of Geography, Maynooth University, Maynooth, Ireland

**Correspondence:** Patrick Pieper (Patrick.Pieper@uni-hamburg.de)

**Abstract.** The Standardized Precipitation Index (SPI) is a widely accepted drought index. Its calculation algorithm normalizes the index via a distribution function. Which distribution function to use is still disputed within the literature. This study illuminates that long-standing dispute and proposes a solution that ensures the normality of the index for all common accumulation periods in observations and simulations.

We compare the normality of SPI time-series derived with the gamma, Weibull, generalized gamma, and the exponentiated Weibull distribution. Our normality comparison is based on a complementary evaluation. Actual against theoretical occurrence probabilities of SPI categories evaluate the absolute performance of candidate distribution functions. Complementary, Akaike's information criterion evaluates candidate distribution functions relative to each other while analytically punishing complexity. SPI time-series, spanning 1983–2013, are calculated from Global Precipitation Climatology Project's monthly precipitation data-set and seasonal precipitation hindcasts from the Max Planck Institute Earth System Model. We evaluate these SPI time-series over the global land area and for each continent individually during winter and summer. While focusing on regional performance disparities between observations and simulations that manifest in an accumulation period of 3-months, we additionally test the drawn conclusions for other common accumulation periods (1-, 6-, 9-, and 12-months).

Our results suggest that calculating SPI with the commonly used gamma distribution leads to deficiencies in the evaluation of ensemble simulations. Replacing it with the exponentiated Weibull distribution reduces the area of those regions, where the index does not have any skill for precipitation obtained from ensemble simulations by more than one magnitude. The exponentiated Weibull distribution maximizes also the normality of SPI obtained from observational data and a single ensemble simulation. We demonstrate that calculating SPI with the exponentiated Weibull distribution delivers better results for each continent and every investigated accumulation period, irrespective of the heritage of the precipitation data. Therefore, we advocate the employment of the exponentiated Weibull distribution as the basis for SPI.

## 1 Introduction

Drought intensity, onset, and duration are commonly assessed with the Standardized Precipitation Index (SPI). SPI was first introduced by McKee et al. (1993) as a temporally and spatially invariant probability-based drought index. In 2011, the World Meteorological Organization (WMO) endorsed the index and recommended its use to all meteorological and hydrological ser-

vices for classifying droughts (Hayes et al., 2011). Advantages of SPI are its standardization (Sienz et al., 2012), its simplicity, and its variable time scale which allows its application to assess meteorological, agricultural, and hydrological drought (Lloyd-Hughes and Saunders, 2002). In contrast, the index's main disadvantage is the mean by which its standardization is realized and concerns the identification of a suitable theoretical distribution function to describe and normalize highly non-normal precipitation distributions (Lloyd-Hughes and Saunders, 2002). The choice of that suitable theoretical distribution function is a

key decision in the index's algorithm (Blain et al., 2018; Stagge et al., 2015; Sienz et al., 2012). This study illuminates reasons for a missing consensus on this choice and attempts to establish such a consensus for both simulations and observations.

SPI quantifies the standardized deficit (or surplus) of precipitation over any period of interest – also called accumulation period. This is achieved by fitting a probability density function (PDF) to the frequency distribution of precipitation totals of the accumulation period – which typically spans either 1-, 3-, 6-, or 12-months. SPI is then generated by applying a Z-

transformation to the probabilities and is standard normal distributed.

The choice of the PDF fitted to the frequency distribution of precipitation is essential because only a proper fit appropriately standardizes the index. While the standardization simplifies further analysis of the SPI, the missing physical understanding of the distribution of precipitation leads to a questionable basis for the fit. Therefore, the choice of the PDF is to some extent arbitrary and depicts the Achilles heel of the index.

Originally, McKee et al. (1993) proposed a simple gamma distribution – while Guttman (1999) identified the Pearson Type III distribution – to best describe observed precipitation. Both of these distributions are nowadays mostly used in SPI's calculation algorithms. As a result, many studies that use SPI directly fit the gamma (Mo and Lyon, 2015; Ma et al., 2015; Yuan and Wood, 2013; Quan et al., 2012; Yoon et al., 2012) or the Pearson type III distribution (Ribeiro and Pires, 2016) without assessing the normality of SPI's resulting distribution with goodness-of-fit tests or other statistical analyses beforehand. The selected

PDF, however, is of critical importance because the choice of this PDF is the key decision involved in the calculation of SPI and indeed many authors have urged to investigate the adequacy of distribution functions for new data-sets and regions before applying them (Blain et al., 2018; Stagge et al., 2015; Touma et al., 2015; Sienz et al., 2012). Neglecting such an investigation has potentially far-reaching consequences in terms of a biased drought description (Guenang et al., 2019; Sienz et al., 2012). A biased drought description would result from an inadequacy of the fitted distribution function to describe precipitation. Such

an inadequacy has been identified for the gamma (Guenang et al., 2019; Blain et al., 2018; Blain and Meschiatti, 2015; Stagge et al., 2015; Sienz et al., 2012; Touma et al., 2015; Naresh Kumar et al., 2009; Lloyd-Hughes and Saunders, 2002) as well as the Pearson type III distribution (Blain et al., 2018; Blain and Meschiatti, 2015; Stagge et al., 2015) in many parts of the world. This lead to the request for further investigations of candidate distribution functions (Blain et al., 2018; Blain and Meschiatti, 2015; Stagge et al., 2015; Touma et al., 2015; Sienz et al., 2012; Lloyd-Hughes and Saunders, 2002; Guttman, 1999).

Several studies have investigated the adequacy of PDFs fitted onto observed precipitation while focusing on different candidate distribution functions (Blain and Meschiatti, 2015), different parameter estimation methods in the fitting procedure (Blain et al., 2018), different SPI time scales (Guenang et al., 2019), general drought climatology (Lloyd-Hughes and Saunders, 2002), and even the most appropriate methodology to test different candidate distribution functions (Stagge et al., 2015).

As each of these investigations analyzed different regions, different PDFs, or focused on different perspectives of this highly multi-dimensional problem, they recommend different candidate PDFs.

Nevertheless, some common conclusions can be drawn. Most investigations only analyzed 2-parameter distribution functions (Guenang et al., 2019; Blain et al., 2018; Stagge et al., 2015; Lloyd-Hughes and Saunders, 2002). Among those, they agreed depending on the accumulation period and/or the location either on the Weibull or the gamma distribution to be best suited in most cases. However, Blain and Meschiatti (2015) also investigated 3-, 4- and 5-parameter distribution functions and concluded that 3-parameter PDFs seem to be best suited to compute SPI in Pelotas, Brazil. Consequently, they advocated for a re-evaluation of the widespread use of the 2-parameter gamma distribution (see also Wu et al., 2007). Moreover, a single candidate distribution function was neither suited in each location nor for each accumulation period to properly calculate SPI time series (Guenang et al., 2019; Blain et al., 2018; Stagge et al., 2015; Lloyd-Hughes and Saunders, 2002). Further, at the accumulation period of 3-months, a critical phase transition in precipitation totals seem to manifest which complicates the overall ranking of candidate PDFs (Guenang et al., 2019; Blain et al., 2018; Stagge et al., 2015). Findings point at the Weibull distribution to be best suited for short accumulation periods (smaller than 3 months) and the gamma distribution for long accumulation periods (larger than 3 months) (Stagge et al., 2015).

Two additional studies analyzed the adequacy of different candidate PDFs fitted onto simulated precipitation while focusing on drought occurrence probabilities in climate projections (Touma et al., 2015; Sienz et al., 2012). Touma et al. (2015) is the only study that tested candidate PDFs globally. However, they solely provide highly aggregated results that are globally averaged for accumulation periods between 3- and 12-months and conclude that the gamma distribution is overall best suited to calculate SPI. In contrast, Sienz et al. (2012) is up to now the only study that tested candidate PDFs in simulations as well as in observations and identified notable differences in their performance in both realizations. They focused on an accumulation period of 1-month and their results also show that the Weibull distribution is well suited for SPI calculations at short accumulation periods in observations but also in simulations. Moreover, their results also hint at the phase transition mentioned above: for accumulation periods longer than 3 months their results indicate that the gamma distribution outperforms the Weibull distribution in observation as well as in simulations. More interestingly, Sienz et al. (2012) results indicate that two 3-parameter distributions (the generalized gamma and the exponentiated Weibull distribution) perform for short accumulation periods as well as the Weibull distribution and for long accumulation periods similar to the gamma distribution; in observations and simulations. Surprisingly, neither the exponentiated Weibull nor the generalized gamma distribution has been thoroughly tested since.

Testing the performance of 3-parameter distributions introduces the risk of overfitting (Stagge et al., 2015; Sienz et al., 2012) which could explain the focus on 2-parameter distributions in recent studies. As a consequence of this one-sided focus in combination with the inability of 2-parameter PDFs to perform sufficiently well in different locations and for different accumulation periods concurrently, many studies have proposed a multi-distribution approach (Guenang et al., 2019; Blain and Meschiatti, 2015; Touma et al., 2015; Sienz et al., 2012; Lloyd-Hughes and Saunders, 2002). Such an approach recommends the use of a set of PDFs. The best-suited PDF of this set is then employed. Thus, the employed PDF might differ depending on the accumulation period, the location, or the data-set. In opposition, other studies have strongly emphasized concern about this

approach, because it adds complexity while reducing or even obliterating comparability across space and time (Stagge et al.,
2015; Guttman, 1999). The comparability across space and time is a main advantage of SPI. Guttman (1999) even warns of
using SPI widely until a single PDF is commonly accepted and established as the norm.

Most studies test candidate distribution functions with goodness-of-fit tests (Guenang et al., 2019; Blain et al., 2018; Blain
and Meschiatti, 2015; Stagge et al., 2015; Touma et al., 2015; Lloyd-Hughes and Saunders, 2002). In this process, some studies
heavily rely on the Kolmogorov-Smirnov test (Guenang et al., 2019; Touma et al., 2015). However, the Kolmogorov-Smirnov
test has an unacceptably high likelihood of erroneously accepting a non-normal distribution if the parameters of the candidate
PDF have been estimated from the same data on which the tested distribution is based (which is because of scarce precipitation
data availability usually always the case) (Blain et al., 2018; Blain and Meschiatti, 2015; Stagge et al., 2015). Therefore, other
studies tested the goodness-of-fit either with an adaptation of the Kolmogorov-Smirnov test, the Lillieforts test (Blain et al.,
2018; Blain and Meschiatti, 2015; Stagge et al., 2015; Lloyd-Hughes and Saunders, 2002), with the Anderson-Darling test
(Blain et al., 2018; Stagge et al., 2015) or with the Shapiro-Wilk test (Blain et al., 2018; Blain and Meschiatti, 2015; Stagge
et al., 2015). Nevertheless, the Lillieforts and Anderson-Darling tests are inferior to the Shapiro-Wilk test (Blain et al., 2018;
Stagge et al., 2015) which in turn is unreliable to evaluate SPI normality (Naresh Kumar et al., 2009).

The above-mentioned goodness-of-fit tests equally evaluate each value of SPI's distribution. Such an evaluation focuses
on the center of the distribution because the center of any distribution contains per definition more samples than the tails. In
contrast, SPI usually analyzes (and thus depends on a proper depiction of) the distribution's tails. Therefore, a blurred focus
manifests in these goodness-of-fit tests. Moreover, the convention to binarily interpret the above-mentioned goodness-of-fit
tests aggravates this blurred focus. Because of this convention, these goodness-of-fit tests are unable to produce any relative
ranking of the performance of distribution functions for a specific location (and accumulation period). This inability prevents
any reasonable aggregation of limitations that surface despite the blurred focus. Thus, they are ill-suited to discriminate the
best performing PDF out of a set of PDFs (Blain et al., 2018). For SPI distributions the question is not whether they are (or
ought to be) normally distributed (for which goodness-of-fit tests are well suited to provide the answer). The crucial question
is rather which PDF maximizes the normality of the resulting SPI distribution. Because of the ill-fitting focus and the ill-suited
convention of these goodness-of-fit tests, they are inept to identify SPI's best-performing candidate distribution function out of
a set of PDFs.

In agreement with this insight, those studies, that rigorously analyzed candidate distribution functions, or investigate an
appropriate test methodology for evaluating SPI candidate PDFs, consequently advocate the use of relative assessments: mean
absolute errors (Blain et al., 2018), Akaike's Information Criterion (AIC) (Stagge et al., 2015; Sienz et al., 2012), or deviations
from expected SPI categories (Sienz et al., 2012). These studies also emphasize the importance of quantifying the differences
between theoretical and calculated SPI values for different drought categories (Blain et al., 2018; Sienz et al., 2012). Stagge
et al. (2015) who investigated appropriate methodologies to test different candidate PDFs even use AIC to discriminate the
performance of different goodness-of-fit tests.

SPI calculation procedures were developed for observed precipitation data. Since models do not exactly reproduce the
observed precipitation distribution, these procedures need to be tested and eventually adapted before being applied to modeled

data. Here, we aspire to identify an SPI calculation algorithm that coherently describes modeled and observed precipitation (i.e. describes both modeled and observed precipitation distributions individually and concurrently). While testing SPI's calculation algorithm on modeled precipitation data is usually neglected, such a test demands nowadays a similarly prominent role as the one for observations because of the increasing importance of drought predictions and their evaluation. Despite this importance, the adequacy of different candidate distribution functions has to the authors' best knowledge never been tested in the output of a seasonal prediction system – although seasonal predictions constitute our most powerful tool to predict individual droughts. To close that gap, this study evaluates the performance of candidate distribution functions in an output of 10 ensemble members of initialized seasonal hindcast simulations.

In this study, we test the adequacy of the gamma, Weibull, generalized gamma, and exponentiated Weibull distribution in SPI's calculation algorithm. The evaluation of their performance depends on the normality of the resulting SPI time-series. In this evaluation, we focus on an SPI accumulation period of 3-months ($\text{SPI}_{3M}$) during winter (DJF) and summer (JJA) and test the drawn conclusions for other common accumulation periods (1-, 6-, 9-, and 12-months). Our analysis conducts two complementary evaluations of their normality: (i) evaluating their normality in absolute terms by comparing actual occurrence probabilities of SPI categories (as defined by WMO's *SPI User Guide* (Svoboda et al., 2012)) against well-known theoretically expected occurrence probabilities from the standard normal distribution ($\mathcal{N}_{0,1}$), (ii) evaluating their normality relative to each other with Akaike's information criterion (AIC) which analytically assesses of the *optimal trade-off* between information gain against the complexity of the PDF to adhere to the risk of overfitting. During this analysis, we investigate observations and simulations. Observed and simulated precipitation is obtained from the monthly precipitation data-set of the Global Precipitation Climatology Project (GPCP) and the above mentioned initialized seasonal hindcast simulations, respectively. We conduct our analysis for the period 1982 to 2013 with a global focus which also highlights regional disparities on every inhabited continent (Africa, Asia, Australia, Europe, North America, and South America).

## 2   Methods

### 2.1   Model and Data

We employ a seasonal prediction system (Baehr et al., 2015; Bunzel et al., 2018) which is based on the Max-Planck-Institute Earth System Model (MPI-ESM). MPI-ESM, also used in the Coupled Model Intercomparison Project 5 (CMIP5), consists of an atmospheric (ECHAM6) (Stevens et al., 2013), and an oceanic (MPIOM) (Jungclaus et al., 2013) component. For this study the model is initialized in May and November and runs with 10 ensemble members in the low-resolution version – MPI-ESM-LR: T63 (approx. 1.875°x1.875°) with 47 different vertical layers in the atmosphere between the surface and 0.01 hPa and GR15 (maximum 1.5°x1.5°) with 40 different vertical layers in the ocean. Except for an extension of the simulation period by 3 years (extended to cover the period 1982–2013), the investigated simulations are identical to the 10-member ensemble simulations analyzed by Bunzel et al. (2018). Here, we analyze the sum of convective and large-scale precipitation from these simulations (Pieper et al., 2020).

We obtain observed precipitation from the Global Precipitation Climatology Project (GPCP) which combines observations and satellite precipitation data into a monthly precipitation data-set on a 2.5°x2.5° global grid spanning 1979 to present (Adler et al., 2003). To compare these observations against our hindcasts, the precipitation output of the model is interpolated to the same grid as GPCP's precipitation data-set from which we only use the simulated time-period (1982–2013).

Depending on the accumulation period (1-, 3-, 6-, 9-, or 12-months) we calculate the frequency distribution of modeled and observed precipitation totals over 2 different seasons (August and February (1), JJA and DJF (3), MAMJJA and SONDJF (6), and so on). Because our results do not indicate major season-dependent differences in the performance of candidate PDFs for $SPI_{3M}$, we aggregate our results for the other accumulation periods over both seasons.

Our precipitation hindcasts are neither bias- nor drift-corrected and also not recalibrated. Such corrections usually adjust the frequency distribution of modeled precipitation in each grid-point to agree better with the observed frequency distribution. Here, we investigate the adequacy of different PDFs in describing the frequency distribution of modeled precipitation totals over each accumulation period without any correction. As a consequence, we require that SPI's calculation algorithm deals with such differing frequency distributions on its own. That requirement enables us to identify the worst possible miss-matches.

## 2.2   Standardized Precipitation Index

We calculate SPI (McKee et al., 1993) for our observed and modeled time-period by fitting a PDF onto sorted 3-months precipitation totals in each grid-point during both seasons of interest and for each accumulation period. Zero-precipitation events are excluded from the precipitation time-series before fitting the PDF and dealt with later specifically. We estimate the parameters of our candidate PDFs in SPI's calculation algorithm with the maximum likelihood method (Nocedal and Wright, 1999) which is also the basis for the AIC computation.

Our parameter estimation method first identifies starting values for the $n$ parameters of the candidate PDFs by roughly scanning the $n$-dimensional phase-space spanned by these parameters. The starting values identified from that scan are optimized with the simulated annealing method (SANN) (Bélisle, 1992). Subsequently, these by SANN optimized starting values are again further optimized by a limited-memory modification of the Broyden-Fletcher-Goldfarb-Shanno (also known as BFGS) quasi-Newton method (Byrd et al., 1995). If the BFGS quasi-Newton method leads to a convergence of the parameters of our candidate PDF, we achieve our goal and end the optimization here. If the BFGS quasi-Newton method does not lead to a convergence of the parameters of our candidate PDF, then we circle back to the starting values optimized by SANN and optimize them again further but this time with the Nelder-Mead method (Nelder and Mead, 1965). After identifying converging parameters, the probabilities of encountering the given precipitation totals are computed and transformed into cumulative probabilities $(G(x))$.

If neither the BFGS quasi-Newton nor the Nelder-Mead method leads to any convergence of the most suitable parameters of our candidate PDFs, then we omit these grid-points where convergence is not achieved. For the gamma, Weibull, and exponentiated Weibull distribution, non-converging parameters are rare exceptions and only occur in a few negligible grid-points. For the generalized gamma distribution, however, non-convergence appears to be a more common issue and occurs

in observations as well as in simulations in roughly every fifth grid-point of the global land area. This shortcoming of the generalized gamma distribution needs to be kept in mind when concluding its potential adequacy in SPI's calculation algorithm.

Since PDFs that describe the frequency distribution of precipitation totals are required to be only defined for the positive real axis, the cumulative probability ($G(x)$) is undefined for $x = 0$. Nevertheless, the time-series of precipitation totals may contain events in which zero precipitation has occurred over the entire accumulation period. Therefore the cumulative probability is adjusted:

$$H(x) = q + (1 - q)G(x) \tag{1}$$

where $q$ is the occurrence probability of zero-precipitation events in the time-series of precipitation totals. $q$ is estimated by the fraction of the omitted zero-precipitation events in our time-series. Next, we calculate from the new cumulative probability ($H(x)$) the likelihood of encountering each precipitation event of our time-series for every grid-point in each season of interest and each accumulation period. In the final step, analog to McKee et al. (1993), a Z-transformation of that likelihood to the standard normal (mean=0, variance=1) variable $Z$ takes place which constitutes the time-series of SPI.

In very arid regions or those with a distinct dry season, SPI time-series are characterized by a lower bound (Pietzsch and Bissolli, 2011; Wu et al., 2007). That lower bound results from $H(x)$ dependence on $q$ and correctly ensures that short periods without rain do not necessarily constitute a drought in these regions. Nevertheless, that lower bound also leads to non-normal distributions of SPI time-series. The shorter the accumulation period, the more likely it is for zero-precipitation events to occur – and the more likely it becomes for SPI time-series to be non-normally distributed. Stagge et al. (2015) proposed to use the *centre of mass* instead of the fraction of zero-precipitation events to estimate $q$. Such an adaptation leads to a lower $q$ than the fraction-approach which distinctly increases the normality of SPI time-series and their statistical interpretability if that fraction becomes larger than approximately one third. As explained before, we want to investigate the worst possible case and, therefore, conservatively estimate $q$. As a consequence, SPI time-series are calculated exclusively for grid-points exhibiting zero-precipitation events in less than 34 % of the times in our time-period. This limitation restricts the SPI calculation in simulations over the Sahara and the Arabian Peninsula for accumulation periods of 1- and 3-months, only exceptionally occurs for an accumulation period of 6-months and does not restrict accumulation periods longer than 6-months. Current complex climate models parameterize convection and cloud micro-physics to simulate precipitation which leads to spurious precipitation amounts. Those spurious precipitation amounts prevent us from directly identifying the probability of zero-precipitation events in modeled precipitation time-series. Analog to Sienz et al. (2012), we prescribe a threshold of 0.035 mm month$^{-1}$ to differentiate between months with and without precipitation in the hindcasts.

To further optimize the fit of the PDF onto modeled precipitation, all hindcast ensemble members are fitted at once. We checked and ascertained the underlying assumption of this procedure – that all ensemble members show in each grid-point identical frequency distributions of precipitation. It is, therefore, reasonable to presume that a better fit is achievable for simulated rather than for observed precipitation.

## 2.3 Candidate Distribution Functions

Cumulative precipitation sums are described by skewed distribution functions which are only defined for the positive real axis. We test four different distribution functions and evaluate their performance based on the normality of their resulting SPI frequency distributions. The four candidate PDFs either consist of a single shape ($\sigma$) and scale ($\gamma$) parameter or include (in the case of the two 3-parameter distributions) a second shape parameter ($\alpha$). Figure 1 displays examples of those four candidate PDFs and their 95 % quantiles for 3-months precipitation totals idealized to be distributed according to the respective distribution function with $\sigma = \gamma = (\alpha) = 2$. Table 1 lists the abbreviations used for these four candidate distribution functions.

Instead of investigating the Pearson Type III distribution, which is already widely used, we analyze the simple gamma distribution. They differ by an additional location parameter which does not change the here presented results (Sienz et al., 2012). Moreover, other studies have demonstrated that the Pearson type III distribution delivers results that are virtually identical to the 2-parameter gamma distribution (Pearson's r = 0.999) (Giddings et al., 2005) and argued that the inclusion of a location parameter unnecessarily complicates the SPI algorithm (Stagge et al., 2015). Therefore, our 3-parameter candidate PDFs comprise a second shape parameter instead of a location parameter. The optimization of this second shape parameter also requires the re-optimization of the first two parameters. The fitting procedure of 3-parameter PDFs needs therefore considerable more computational resources than the fitting procedure of 2-parameter distribution functions.

1. Gamma distribution

$$f(x) = \frac{1}{\sigma\Gamma(\gamma)} \left(\frac{x}{\sigma}\right)^{\gamma-1} \exp(-\frac{x}{\sigma}) \tag{2}$$

The gamma distribution ($\Gamma$ being the gamma-function) is typically used for SPI calculations directly or in its location parameter extended version: the Pearson Type III distribution (Guttman, 1999). The results of the gamma distribution also serve as proxy for the performance of the Pearson Type III distribution.

2. Weibull distribution

$$f(x) = \frac{\gamma}{\sigma} \left(\frac{x}{\sigma}\right)^{\gamma-1} \exp(-\left(\frac{x}{\sigma}\right)^{\gamma}) \tag{3}$$

The Weibull distribution is usually used to characterize wind speed. Several studies identified the Weibull distribution, however, to perform well in SPI's calculation algorithm for short accumulation periods (Guenang et al., 2019; Blain et al., 2018; Stagge et al., 2015; Sienz et al., 2012).

3. Generalized gamma distribution

$$f(x) = \frac{\alpha}{\sigma\Gamma(\gamma)} \left(\frac{x}{\sigma}\right)^{\alpha\gamma-1} \exp(-\left(\frac{x}{\sigma}\right)^{\alpha}) \tag{4}$$

The generalized gamma distribution extends the gamma distribution by another shape-parameter ($\alpha$). In the special case of $\alpha = 1$ the generalized gamma distribution becomes the gamma distribution and for the other special case of $\gamma = 1$ the generalized gamma distribution becomes the Weibull distribution. Sienz et al. (2012) identified the generalized gamma distribution as promising candidate distribution function for SPI's calculation algorithm.

4. Exponentiated Weibull distribution

$$f(x) = \frac{\alpha\gamma}{\sigma} \left(\frac{x}{\sigma}\right)^{\gamma-1} \left[1 - \exp\left(-\left(\frac{x}{\sigma}\right)^{\gamma}\right)\right]^{\alpha-1} \tag{5}$$

The exponentiated Weibull distribution extends the Weibull distribution by a second shape parameter ($\alpha$). For $\alpha = 1$ the exponentiated Weibull distribution becomes the Weibull distribution. Sienz et al. (2012) revealed that the exponentiated Weibull distribution performs well in SPI's calculation algorithm.

## 2.4 Deviations from the Standard Normal Distribution

SPI time-series are supposed to be standard normally distributed ($\mu = 0$ and $\sigma = 1$). Thus, we evaluate the performance of each candidate distribution function (in describing precipitation totals) based on the normality of their resulting SPI frequency distributions. In this analysis, we calculate actual occurrence probabilities for certain ranges of events in our SPI frequency distributions and compare those actual against well-known theoretical occurrence probabilities for the same range of events. We then evaluate the performance of each candidate distribution function and their resulting SPI time-series based on the magnitude of deviations from the standard normal distribution ($\mathcal{N}_{0,1}$). These deviations are henceforth referred to as deviations from $\mathcal{N}_{0,1}$.

According to WMO's *SPI User Guide* (Svoboda et al., 2012) (see Table 2), SPI distinguishes between seven different SPI categories. These seven different categories with their pre-defined SPI intervals serve as analyzed ranges of possible events in our analysis. It is noteworthy here, that these seven SPI categories differ in their occurrence probabilities. The occurrence of normal conditions (N0) is more than twice as likely than all other six conditions put together. Therefore, any strict normality analysis of SPI time-series would weigh each classes' identified deviation from $\mathcal{N}_{0,1}$ with the occurrence probability of the respective class. However, when analyzing droughts with SPI, one is usually interested in extreme precipitation events. Thus, it seems less important for the center of SPI's distribution to be normally distributed. Instead, it is intuitively particularly important for the tails (especially the left-hand tail) of the distribution to adhere to the normal-distribution. The better the tails of our candidate PDF's SPI distributions agree with $\mathcal{N}_{0,1}$, the better is our candidate PDF's theoretical description of extreme precipitation events. For this reason, we treat all seven SPI categories equally, irrespective of their theoretical occurrence probability.

The 3-parameter candidate distribution functions contain the 2-parameter candidate distribution functions for special cases. Given those special cases, the 3-parameter candidate distribution functions will in theory never be inferior to the 2-parameter candidate distribution functions they contain when analyzing deviations from $\mathcal{N}_{0,1}$ – assuming a sufficient quantity of input data which would lead to a sufficient quality of our fit. Thus, the question is rather whether deviations from $\mathcal{N}_{0,1}$ reduce enough to justify the 3-parameter candidate distribution functions' requirement of an additional parameter. An additional parameter that needs to be fitted increases the risk of overfitting (Stagge et al., 2015; Sienz et al., 2012). On the one hand, the final decision on this trade-off might be subjective and influenced by computational resources available or by the length of the time-series which is to be analyzed because fitting more parameters requires more information. Moreover, it might well be wiser to employ scarce computational resources in optimizing the fit rather than increasing the complexity of the PDF. On the other

hand, assuming computational resources and data availability to be of minor concern, there exists an analytical way to tackle this trade-off: Akaike's Information Criterion (Akaike, 1974).

## 2.5 Akaike's Information Criterion

Our aim is twofold. First, we want to maximize the normality of our SPI time-series by choosing an appropriate distribution function. Second, we simultaneously aspire to minimize the parameter-count of the distribution function to avoid unnecessary complexity. Avoiding unnecessary complexity decreases the risk of overfitting. The objective is to identify the necessary (minimal) complexity of the PDF which prevents the PDF from being too simple and lose explanatory power. Or in other words: we are interested in the so-called *optimal trade-off* between bias (PDF is too simple) and variance (PDF is too complex). Akaike's information criterion (AIC) performs this trade-off analytically (Akaike, 1974). AIC estimates the value of information gain (acquiring an improved fit) and penalizes complexity (the parameter count) directly by estimating the Kullback-Leibler information (Kullback and Leibler, 1951):

$$AIC = -2\ln \mathcal{L}(\hat{\theta}|y) + 2k \tag{6}$$

$\mathcal{L}(\hat{\theta}|y)$ describes the likelihood of specific model-parameters ($\hat{\theta}$) with given data from which these parameters were estimated ($y$). $k$ describes the degrees of freedom of the candidate PDF (the parameter-count which equates dependent on the candidate PDF either to 2 or 3). Analogue to Burnham and Anderson (2002), we modified the last term from $2k$ to $2k + (2k(k+1))/(n - k - 1)$ in order to improve the AIC calculation for small sample sizes ($n/k < 40$), whereas in our case $n$ corresponds to the sample size of the examined period (31 for observations and 310 for simulations). The modified version approaches the standard version for large $n$.

In our case, AIC's first term evaluates the performance of candidate PDFs in describing the given frequency distributions of precipitation totals. The second term penalizes candidate PDFs based on their parameter-count. The best-performing distribution function attains the smallest AIC value because the first term is negative and the second one is positive.

Further, the absolute AIC value is often of little information – especially in contrast to relative differences between AIC values derived from different distribution functions. Thus, we use relative AIC differences (AIC-D) in our analysis. We calculate these AIC-D values for each PDF by computing the difference between its AIC value to the lowest AIC value of all four distribution functions. AIC-D values inform us about superiority in the optimal trade-off between bias and variance and are calculated as follows:

$$AIC\text{-}D_i = AIC_i - AIC_{min} \tag{7}$$

The index $i$ indicates different distribution functions. $AIC_{min}$ denotes the AIC value of the best-performing distribution function.

For our analysis, AIC-D values are well suited to compare and rank different candidate PDFs based on their trade-off between bias and variance. The best performing distribution function is characterized by a minimum AIC value ($AIC_{min}$) which translates to an AIC-D value of $0$. It seems noteworthy here that any evaluation of (or even any discrimination between)

candidate distribution functions, which exhibit sufficiently similar AIC-D values, is unfeasible as a consequence of our rather small sample size (particularly in observations, but also in simulations). AIC-D values below two ought to be in general interpreted as an indicator of substantial confidence in the performance of the model (here, the PDF). In contrast, AIC-D

values between four and seven indicate considerably less confidence and values beyond ten essentially none (Burnham and Anderson, 2002).

The analysis of deviations from $\mathcal{N}_{0,1}$ assesses performances of candidate PDFs in absolute terms irrespective of the candidate PDF's complexity. In contrast, the AIC-D analysis evaluates the performance of candidate PDFs relative to each other while analytically punishing complexity. Consequently, the AIC-D analysis cannot evaluate whether the best-performing candidate

distribution function also performs adequately in absolute terms. In opposition, deviations from $\mathcal{N}_{0,1}$ encounter difficulties when evaluating whether an increased complexity from one PDF to another justifies any given improvement. Both analyses together, however, augment each other complementary. This enables us to conclusively investigate: (i) which candidate PDF performs best while (ii) ensuring adequate absolute performance and while (iii) constraining the risk of over-fitting.

## 2.6   Aggregation of Results over Domains

For each candidate distribution function, accumulation period, domain, and during both seasons, we compute deviations from $\mathcal{N}_{0,1}$ separately for observations and simulations as schematically depicted on the left-hand side in Fig. 2. First, we count the events of each SPI category in every land grid-point globally. For each category, we then sum the category counts over all grid-points that belong to the domain of interest. Next, we calculate actual occurrence probabilities through dividing that sum by the sum over the counts of all seven SPI categories (per grid-point there are 31 total events in observations and 310 in

simulations). In a final step, we compute the difference to theoretical occurrence probabilities of $\mathcal{N}_{0,1}$ (provided in Table 2) for each SPI category and normalize that difference – expressing the deviation from $\mathcal{N}_{0,1}$ as a percentage of the theoretically expected occurrence probability.

Again for each candidate distribution function, accumulation period, domain, and both seasons, we aggregate AIC-D over several grid-points into a single graph separately for observations and simulations as depicted on the right-hand side of the flow

chart in Fig. 2. For each domain, we compute the fraction of total grid-points of that domain for which each candidate PDF displays an AIC-D value equal to or below a specific AIC-$D_{max}$ value. That calculation is iteratively repeated for infinitesimally increasing AIC-$D_{max}$ values. In this representation, the probabilities of all PDFs, at the specific AIC-$D_{max}$ value of 0, sum up to 100 % because only one candidate PDF can perform best in each grid-point. Thus, we arrive at a summarized AIC-D presentation in which those candidate distribution functions which approach 100 % the fastest (preferably before the specific

AIC-$D_{max}$ value of 4; ideally even before the AIC-$D_{max}$ value of 2) are better suited than the others.

## 2.7   Regions

We investigate the normality of SPI time-series derived from each candidate PDF first for the entire global land area and analyze subsequently region-specific disparities. For this analysis we focus on the land area over six regions scattered over all six inhabited continents: Africa (0°–30°S; 10°E–40°E), Asia (63°N–31°N; 86°E–141°E), Australia (16°S–38°S; 111°E–

153°E), Europe (72°N–36°N; 10°W–50°E), North America (50°N–30°N; 130°W–70°W), and South America (10°N–30°S; 80°W–35°E) (Fig. 3).

Examining frequency distributions of precipitation totals over smaller domains than the entire globe reduces the risk of encountering opposite deviations from $\mathcal{N}_{0,1}$ for the same category that balance each other in different grid-points with unrelated climatic characteristics. This statement is based on either one of the following two assumptions. First, the sum over fewer grid-

points is less likely to produce deviations which balance each other. Second, the frequency distribution of precipitation totals is likely to be more uniform for grid-points that belong to the same region (and therefore exhibit similar climatic conditions) than when they are accumulated over the entire globe. One could continue along this line of reasoning because the smaller the area of the analyzed regions, the more impactful are both of these assumptions. However, comparing actual against theoretically expected occurrence probabilities with a scarce database (31 events in observations) will inevitably produce deviations. In

observations, we would expect in each grid-point that 0.7 extremely wet/dry and 1.4 severely wet/dry events occur over 31 years. Thus, deviations in different grid-points need to balance each other to some extent, to statistically evaluate and properly compare candidate PDFs. The crucial performance requirement demands that they balance each other also when averaged over sufficiently small domains with similar climatic conditions.

For a first overview, it is beneficial to cluster as many similar results as possible together to minimize the level of complexity

of the regional dimension. The choice of sufficiently large/small domains is still rather subjective. Which size of regions is most appropriate? This subjective nature becomes apparent in studies that identify differing borders for regions that are supposed to exhibit rather uniform climatic conditions (Giorgi and Francisco, 2000; Field et al., 2012). Instead of using *Giorgi-Regions* (Giorgi and Francisco, 2000) or *SREX-Regions* (Field et al., 2012), we opt here for a broader and more continental picture.

## 3   Results

### 3.1   SPI Accumulation Period of 3-Month

#### 3.1.1   Global

In agreement with prior studies (Blain et al., 2018; Lloyd-Hughes and Saunders, 2002; McKee et al., 1993), the 2-parameter gamma distribution (GD2) describes on the global average the observed frequency distribution of $SPI_{3M}$ rather well during the boreal winter (DJF) and summer (JJA) (Fig. 4, (a)). Contrary to Sienz et al. (2012), who investigated $SPI_{1M}$ time-series, the 2-

parameter Weibull distribution (WD2) delivers a poor frequency distribution of $SPI_{3M}$ during both seasons (Fig. 4, (b)). Aside from GD2, GGD3 and EWD3 also perform adequately in absolute terms for observations. Discriminating their deviations from $\mathcal{N}_{0,1}$ is difficult. On the one hand, GD2 represents the especially important left-hand tail of $SPI_{3M}$ time-series' frequency distribution (D3) in JJA worse than our 3-parameter candidate PDFs (compare Fig. 4, (a) against (c) and (d)). On the other hand, GD2 displays smaller deviations from $\mathcal{N}_{0,1}$ than our 3-parameter candidate PDFs in the center of the SPI's distribution.

Despite these minor differences, and in agreement with Sienz et al. (2012), GGD3 and EWD3 perform overall similar to GD2 (compare Fig. 4, (a) against (c) and (d)).

In theory, since the 3-parameter generalized gamma distribution (GGD3) encompasses GD2 as a special case, GGD3 should not be inferior to GD2. In reality, however, the applied optimization methods appear to be too coarse for GGD3 to always lead to an identical or better optimum than the one identified for GD2 with the given length of the time-series. When optimizing 3 parameters it is more likely to miss a specific constellation of parameters which would further optimize the fit; especially when limited computational resources impede the identification of the actual optimal fitting parameters. Additionally, a limited database (our database spans 31 years) obscures the frequency distribution of precipitation totals which poses another obstacle to the fitting methods. This results in missed optimizations opportunities that impact GGD3 stronger than GD2 because of GGD3's increased complexity which leads to GGD3 requiring more data than GD2. Therefore, the weighted sum (weighted by the theoretical occurrence probability of the respective SPI class (Table 2)) over the absolute values of deviations from $\mathcal{N}_{0,1}$ along all SPI categories is lowest for GD2 in both analyzed seasons (see legend in Fig. 4, (a)–(d)).

In agreement with Sienz et al. (2012), who identified notable differences in the performance of candidate PDFs between observations and simulations, this general ranking changes when we consider modeled instead of observed $SPI_{3M}$ time-series (Fig. 4, (e)–(h)). While GD2, GGD3, and EWD3 display similar deviations from $\mathcal{N}_{0,1}$ in observations (Fig. 4 (a), (c), and (d)), a noticeable difference emerges in ensemble simulations (Fig. 4 (e), (g), and (h)). GD2's performs distinctly worse than our 3-parameter PDFs in ensemble simulations.

In simulations, the fit onto 3-months precipitation totals is performed on all ten ensemble members at once. This 10-folds the sample size in simulations relative to observations. Presuming an imperfect fit for the 31 samples in observations, deviations from $\mathcal{N}_{0,1}$ are expected to reduce along our four candidate distribution functions as a result of 10-folding the sample size of their fit. Yet, GD2 does not benefit from 10-folding the sample size. GD2 performs similarly in observations and simulations (Fig. 4 (a) and (e)). In contrast, our 3-parameter PDFs display considerably smaller deviations from $\mathcal{N}_{0,1}$ in ensemble simulations than in observations (compare Fig. 4 (c) and (d) against (g) and (h)). Consequently, both 3-parameter candidate PDFs excel during both seasons in ensemble simulations (Fig. 4, (g) and (h)), while any distinction between both 3-parameter candidate distribution functions is still difficult. On the one side, different frequency distributions between observed and modeled precipitation totals might be one reason for this difference. On the other side, the fit of three parameters also requires more data than the fit of two. It is therefore sensible to expect that 3-parameter PDFs benefit stronger than 2-parameter PDFs from an increase in sample size. Are our 3-parameter candidate PDFs are better suited than our 2-parameter PDFs to describe modeled precipitation distributions? Or benefit our 3-parameter PDFs just stronger than 2-parameter PDFs from an increasing sample size?

We attempt to disentangle both effects (analyzing modeled, instead of observed, precipitation distributions, and increasing the sample size) for our 2-parameter candidate PDFs, next. If the 2-parameter PDFs are suited to be applied to modeled precipitation data, they should benefit at least to some extent from this multiplication of sample size. Despite expecting irregularities in the magnitude of these reductions, they ought to be notable for candidate distribution functions that are adequately suited to describe modeled 3-months precipitation totals – assuming an imperfect fit for the 31 events spanning our observational time-series. Therefore, we weigh each class' deviation from $\mathcal{N}_{0,1}$ by the theoretical occurrence probability (see Table 2) of the respective class and analyze weighted deviations from $\mathcal{N}_{0,1}$.

For the 2-parameter PDFs, the weighted deviations from $\mathcal{N}_{0,1}$ (shown in the legend of Fig. 4) either stay constant (for GD2 in DJF) or increase in simulations relative to observations (compare the legends in the left against the one in the right column of Fig. 4). Relative to observations, GD2's weighted deviations increase in simulations by more than 120% in JJA, while WD2's increase by more than 25% in JJA and 80% in DJF. The most plausible explanation for these weighted deviations to increase, when 10-folding the database, are different frequency distributions between observed and modeled 3-months precipitation totals. Our 2-parameter candidate PDFs are better suited to describe observed than modeled 3-months precipitation totals. In contrast, for our 3-parameter candidate distribution functions, weighted deviations from $\mathcal{N}_{0,1}$ are substantially larger in observations than in simulations. GGD3's (EWD3's) are larger by 210% (500%) and 58% (200%) during DJF and JJA, respectively. The 3-parameter candidate distribution functions benefit strongly from the artificial increase of our time-series and seem better suited than our 2-parameter candidate PDFs to describe precipitation distributions obtained from ensemble simulations.

In this section, we have analyzed global deviations from $\mathcal{N}_{0,1}$ thus far and identified:

– GD2, GGD3, and EWD3 describe similarly well the overall frequency distribution of observed 3-months precipitation totals.

– WD2 performs overall poorly and is in every regard inferior to any other candidate distribution function.

– GGD3 and EWD3 describe the frequency distribution of modeled 3-months precipitation totals distinctly better than any 2-parameter candidate distribution.

– GD2 describes the frequency distribution of modeled 3-months precipitation totals sufficiently well on the global average.

– Both 2-parameter candidate distribution functions are unable to benefit from the increased length of the database in simulations relative to observations, while both 3-parameter PDFs strongly benefit from that increase.

It is noteworthy, that investigating deviations from $\mathcal{N}_{0,1}$ over the entire globe contains the risk of encountering deviations that balance each other in different grid-points with unrelated climatic characteristics. Until dealing with this risk, our analysis of deviations from $\mathcal{N}_{0,1}$ only indicates that three candidate PDFs (GD2, GGD3, and EWD3) display an adequate absolute performance. On the one hand, we can reduce that risk by analyzing deviations from $\mathcal{N}_{0,1}$ only over specific regions. This analysis safeguards our investigation by ensuring (rather than just indicating) an adequate absolute performance around the globe and is performed later. On the other hand, we first completely eliminate this risk by examining AIC-D frequencies: aggregating AIC-D values over the entire globe evaluates the performance of PDFs in each grid-point and normalizes these evaluations by (rather than adding them over) the total number of grid-points of the entire globe. We investigate AIC-D frequencies first to evaluate whether GGD3 and/or EWD3 perform sufficiently better than GD2 to justify their increased complexities.

In general, each candidate distribution function performs similarly well in winter and summer in their depiction of the frequency distribution of observed 3-months precipitation totals (compare Fig. 5, (a) against (b)). In agreement with our previous results and prior studies (Blain et al., 2018; Lloyd-Hughes and Saunders, 2002; McKee et al., 1993), GD2 ideally describes

observed 3-months precipitation totals during both seasons in many grid-points of the global land area (Fig. 5, (a) and (b)).

GD2 displays AIC-D values of less than 2 in approximately 84.5% (83.5%) of the global land area in DJF (JJA). That ought to be interpreted as substantial confidence in GD2's performance in these grid-points. However, beyond an AIC-D$_{max}$ value of 2, EWD3 (and GGD3) approach 100 % coverage considerably faster than GD2. EWD3 quickly compensates for AIC's complexity punishment (which is 2.46 units larger for EWD3 than for GD2 (indicated by the vertical black line in Fig. 5)). Beyond this vertical black line, EWD3 conclusively outperforms GD2 (the only intersection of the yellowish, and the bluish lines coincide with the intersection of that vertical black line in Fig. 5, (a) and (b)). EWD3 performs well (AIC-D$_{max}$ < 4) in virtually every global land grid-point. During DJF (JJA), EWD3 displays globally (in all land grid-points) AIC-D values of less than 5.03 (7.03). In contrast, GD2 performs erroneously (apparent by AIC-D$_{max}$ values in excess of 4) in approximately 7% (6%) of the global land grid-points during DJF (JJA). Further, GD2 performs during both seasons insufficiently (AIC-D$_{max}$ values beyond 7) in 2% and without skill (AIC-D$_{max}$ values beyond 10) in 1% of the global land area. While EWD3 strictly outperforms GGD3, GGD3 still performs similarly to EWD3 in observations. Thus, our focus on EWD3 becomes only plausible during the investigation of AIC-D frequencies in ensemble simulations.

In ensemble simulations, our results are again rather stable for all investigated distribution functions between summer and winter (compare Fig. 5, (c) against (d)). All distribution functions display in both seasons the same distinct ranking of their performance for AIC-D$_{max}$ values of 2 and beyond. EWD3 outperforms GGD3 which is better than GD2, while WD2 performs especially poor. The confidence in GD2 drastically diminishes further when we analyze the performance of our four candidate PDFs in ensemble simulations. EWD3 is superior to any other distribution function in JJA and DJF for each AIC-D$_{max}$ value beyond 1.52 in DJF and 0.73 in JJA (see intersect between yellowish and blueish lines in Fig. 5, (c) and (d)). Assuming those AIC-D$_{max}$ values to be sufficiently small (AIC-D values of less than 2 are practically indistinguishable from each other in their performance), EWD3 performs best among all candidate PDFs in general. We interpret EWD3's performance in ensemble simulations as ideal in approximately 85% (86%) of the global land area during DJF (JJA). For AIC-D$_{max}$ values beyond 2, EWD3 quickly approaches 100 % coverage, again, and performs erroneously or insufficiently only in 1% of the global land area during both seasons. In contrast, GD2 performs erroneously in 23% (30%) and insufficient in 14% (21%) of the global land grid-points during DJF (JJA). Yet, most telling might be the fraction of grid-points in which the candidate PDFs display AIC-D values of 10 and beyond and thus show no skill in ensemble simulations. GD2 fails during DJF (JJA) in 10% (15%) of the global land area. In opposition, EWD3 only fails in 0.45% (0.87%) during DJF (JJA). Ergo, employing EWD3, instead of GD2, reduces the count of grid-points without any skillful performance by over one magnitude (by a factor of roughly 20). EWD3 also universally outperforms GGD3. Given their equal parameter-count, it seems rational to rather employ EWD3 than GGD3.

Analyzing AIC-D frequencies for both seasons (DJF and JJA) discloses no distinct season-dependent differences, similar to before in the investigation of deviations from $\mathcal{N}_{0,1}$. Therefore, we average identified land area coverages over both seasons in the summary of AIC-D frequencies. Table 3 summarizes our findings from the investigation of AIC-D values over the entire global land area during both seasons. EWD3 performs well (AIC-D $\leq$ 4) with substantial confidence (at least 95% of land grid-points conform performance) around the globe in both realizations. Additionally, EWD3 also performs best in each of

these analyses (each row of Table 3 in which we consider its performance with substantial confidence). The other analyzed candidate PDFs perform substantially worse than EWD3 in ensemble simulations and slightly worse in observations.

It seems worth elaborating on the insufficient (only average) confidence in EWD3 to perform ideally in observations (ensemble simulations) around the globe. The complexity penalty of AIC correctly punishes EWD3 stronger than GD2 because AIC evaluates whether EWD3's increased complexity (relative to GD2) is necessary. However, the results justify the necessity for this increased complexity – GD2 performs erroneously in 26% (6%), insufficiently in 18% (2%), and without any skill in 12% (1%) of the global land area in ensemble simulations (observations). The risk of underfitting by using 2-parameter PDFs seems larger than the risk of overfitting by using 3-parameter PDFs. Once the need for 3-parameter candidate PDFs is established, their remaining punishment relative to 2-parameter PDFs biases the analysis; particularly for the ideal AIC-D category. EWD3's increased complexity penalty relative to 2-parameter candidate PDFs depends on the sample size and amounts to 2.46 in observations and 2.04 in ensemble simulations (see black vertical lines in Fig. 5 (a)–(d)). The AIC-D$_{max}$ value beyond which EWD3 reaches coverages close to 100% approximately amounts to EWD3's increased penalty (see Fig. 5 (a)–(d)). Correcting EWD3's coverages for this bias would affect our evaluation of EWD3's performance only for the ideal AIC-D category. To illustrate this effect, we only consider AIC's estimated likelihood (without its penalty). Such a consideration corrects this complexity bias in EWD3's performance. While we analytically analyzed this consideration, a first-order approximation suffices for the scope of this publication. In that first-order approximation of this consideration, we simply shift the curve of EWD3 by 2.46 units leftwards in observations (Fig. 5 (a) and (b))) and by 2.04 units leftwards in ensemble simulations (Fig. 5 (c) and (d)). After this shift, EWD3 would also perform ideal with substantial confidence.

The AIC-D frequencies of Table 3 are robust in all investigated regions except Australia (not shown). In Australia, GD2's performance slightly improves relative to the global results during DJF in observations. In contrast, GD2 performs worse than any other investigated candidate PDFs (even worse than WD2) during JJA in observations and during DJF in simulations. Since these are the only minor regional particularities evident in regional AIC-D frequencies, we will during the regional focus in the remaining analysis of SPI$_{3M}$ solely display, explain, and concentrate on deviations from $\mathcal{N}_{0,1}$.

Among our candidate PDFs, EWD3 is obviously the best-suited PDF for SPI. Yet, we still need to confirm whether also EWD3's absolute performance is adequate. While the global analysis indicated EWD3's adequateness, the ultimate validation of this claim is incumbent upon the regional analysis.

### 3.1.2 Regional Deviations from $\mathcal{N}_{0,1}$

We investigated thus far deviations from $\mathcal{N}_{0,1}$ for the entire global land area. In this process, our results indicate an adequate absolute performance of GD2, GGD2, and EWD3. However, that investigation might be blurred by deviations which balance each other over totally different regions with unrelated climatic characteristics. Thus, we will reduce the area analyzed in this subsection and perform a further aggregated investigation that focuses on each continental region individually. That further aggregation of results dismisses the dimension of different SPI categories because their analysis revealed a rather uniform relation over each region: extreme SPI categories show the largest deviations, while normal conditions exhibit the smallest. As a consequence, we display from now on only unweighted sums over the absolute values of these deviations across all SPI

categories. To provide a more intuitive number for these unweighted sums, we normalize them by our SPI category count (7). Consequently, our analysis will investigate the mean deviations per SPI category, henceforth.

In observations (Fig 6. (a) and (b)), WD2 performs in all analyzed regions again worst of all candidate PDFs in delivering a proper frequency distribution of $SPI_{3M}$ during both investigated seasons. Over all analyzed regions and seasons, EWD3 displays the smallest deviations from $\mathcal{N}_{0,1}$, while GD2 and GGD3 perform only slightly worse. Some minor region-dependent differences emerge. E.g. in Africa, a distinct ranking of the performance of all four candidate distribution functions emerges during JJA – EWD3 outperforms GGD3 which performs better than GD2. Aside, all candidate PDFs display almost identical

deviations from $\mathcal{N}_{0,1}$ over Australia during DJF in observation.

     In simulations (Fig 6. (c) and (d)), the ranking of the performance of different PDFs becomes more distinct than it is in observations during both analyzed seasons and investigated domains, except Australia. This compared to observations easier distinction over almost every region of the globe results from increased mean deviations for GD2, while they stay comparable low for GGD3 and EWD3, relative to the global analysis. As shown before, 2-parameter PDFs ineptly describe precipitation

totals obtained from ensemble simulations. Consequently, during both seasons, GGD3 and EWD3 perform in each region exceptionally well, while GD2 performs overall average at best, whereas WD2 performs still poor in general. The performances of GD2 and WD2 are only in Africa during DJF equally poor which impedes any clear ranking. Similarly difficult is any distinction of their performance in North America during JJA as a consequence of one of WD2's best performances (as also identified by Sienz et al. (2012) for $SPI_{1M}$). Furthermore poses Australia an exception to the identified ranking pattern of

candidate PDFs for simulations. During the austral summer (DJF), WD2 distinctly outperforms GD2 which exhibits the largest mean deviations. Interestingly, analog to the performance of candidate PDFs over Australia in observations during DJF, we identify over Australia also in simulations a season when the performance of all four candidate distribution functions is rather similar. However, this occurs in simulations during JJA.

     These insights about the candidate PDFs performance in observations and simulations are even more obvious at first glance

when displayed in an image plot (Fig. 7 (a) and (b)). The poor performance of WD2 in observations and simulations is obvious over all domains and in both investigated seasons. Also, the exception to this pattern for Australia during the austral summer (Fig. 7 (a)) in simulations is distinctly visible. Evident are further the overall similar performances of GD2, GGD3, and EWD3 in observations over all domains and both analyzed seasons. Further, the generally improved performance of 3-parameter candidate distribution functions (GGD3 and EWD3) relative to 2-parameter candidate PDFs in simulations is

distinctly palpable. Aside, even the better performance of EWD3 relative to GGD3 in Africa generally or in observations over Europe is easily discernible.

     For observations, the regional analysis confirms the insights from the global analysis in each region: EWD3 is (same as GD2 and GGD3) an adequate PDF in SPI's calculation algorithm. For ensemble simulations, the regional analysis additionally corroborates the finding of the AIC-D analysis that EWD3 performs noticeably better than GD2. The corroboration of this

finding substantiates support for EWD3.

The analysis of AIC-D frequencies proves that EWD3 is SPI's best distribution function among our candidate PDFs. Additionally, the regional investigation confirms the global analysis: the absolute performance of EWD3 is at minimum adequate in observations and ensemble simulations.

### 3.1.3 Improvement relative to a multi-PDF Approach and a Baseline

In the following, we investigate deviations from $\mathcal{N}_{0,1}$ for a multi-PDF SPI calculation algorithm which uses in each grid-point that distribution function which yields for this respective grid-point the minimum AIC value (whose AIC-D value equates to 0). An analog SPI calculation algorithm has been repeatedly proposed in literature (Guenang et al., 2019; Blain and Meschiatti, 2015; Touma et al., 2015; Sienz et al., 2012; Lloyd-Hughes and Saunders, 2002). We analyze the impact of such an SPI calculation algorithm and compare those results against a baseline comparison and against the most suitable calculation algorithm

identified in this study which uses EWD3 as PDF. The results obtained from the SPI calculation algorithm that uses a multi-PDF approach are labeled $AIC_{min}$-analysis. As a baseline comparison, we choose the calculation algorithm and optimization method of the frequently used R-package from Beguería and Vicente-Serrano (2017) and refer to these results as baseline. To maximize the comparability of SPI time-series calculated with this baseline, we employ the simple 2-parameter gamma distribution as a calculation algorithm and estimate the parameters of the PDF again with the *maximum-likelihood method*. It

seems noteworthy that our parameter estimation method takes about 60 times longer to find optimal parameters of GD2 than the baseline. The comparison between the performance of our baseline against GD2's performance (compare Fig. 8 against Fig. 7) thus also indicates the impact of the meticulousness applied to the optimization of the same parameter estimation method.

The $AIC_{min}$-analysis performs generally almost identical to EWD3 over each domain and in both realizations (observations and simulations). Further, deviations are not necessarily minimal when computing SPI with the $AIC_{min}$-analysis (Fig. 8, (a)

and (b)). This results from the dependence of AIC's punishment on the parameter count of the distribution function. It is simply not sufficient for EWD3 to perform best by a small margin in order to yield a lower AIC value than GD2/WD2. EWD3 needs to perform sufficiently better to over-compensate its by AIC imposed punishment. Or in other words, EWD3 is expected to perform distinctly better than GD2/WD2 because of its increased complexity. As a consequence, EWD3 is only selected by AIC as the best performing distribution function if it fulfills that expectation.

In contrast to previous results in this and other studies (Stagge et al., 2015), which showed no seasonal differences in the performance of candidate PDFs, the baseline performs overall better in JJA than in DJF (compare in Fig. 8, (a) against (b)). Relative to our findings in the previous subsection (Fig 7.), the baseline performs similar to GD2 in JJA but worse than WD2 in DJF (compare Fig. 7 against Fig. 8,). This reveals a substantial impact of the optimization procedure, at least for DJF-precipitation totals. Further, the baseline performs especially poor in describing the frequency distribution of $SPI_{3M}$ in

simulations during the austral summer. It is important to note that the baseline over-estimates modeled extreme droughts during DJF over Australia by more than 240% (not shown). That is by a huge margin the largest deviation we encountered during our analysis and highly undesirable when analyzing droughts. Contrary to Blain et al. (2018), who investigated the influence of different parameter estimation methods on SPI's normality and identified only barely visible effects, the massive difference between the baseline and GD2 in DJF is severely concerning; especially given that the here used parameter estimation methods

are identical and the only difference is the meticulousness of the optimization procedure. Since GD2 and the baseline both use the maximum likelihood method to estimate the PDF's parameters, main differences do not only emerge when using different estimation methods but rather manifest already in the applied procedures by which these methods are optimized.

Unsurprisingly the same deficit as identified before for both 2-parameter candidate PDFs also emerges in the baseline's performance: the by each classes' likelihood of occurrence weighted sum over the absolute values of deviations from $\mathcal{N}_{0,1}$ increases as a result of 10-folding our database (not shown). Although the baseline already performs especially poorly when analyzing weighted deviations during DJF in observations, it performs even worse in simulations; although the performance deteriorates only marginally. Such an increase of weighted deviations is a strong indicator of the baseline's difficulties to sufficiently describe the frequency distribution of modeled $SPI_{3M}$. In the baseline, these weighted deviations increase globally by 2 % in DJF and 40 % in JJA (as a reminder: the weighted deviations stay constant for GD2 in DJF and increase by more than 120 % in JJA). In contrast, these weighted deviations decrease for the $AIC_{min}$-analysis by 70% in DJF and by 60% in JJA around the entire globe (not shown).

Moreover, identifying the maximum deviation from $\mathcal{N}_{0,1}$ for 196 different analyses which range across each SPI category (7), domain (7), both seasons (2), as well as differentiating between observation and simulation (2) (not shown), the baseline performs worst in 79 out of those 196 analyses, while WD2 performs worst in 103 of these analyses. It is noteworthy that out of those 79 analyses in which the baseline performs worst, 63 analyses occur during DJF. As a side note, GD2 performs worst six times with our optimization, while GGD3 and EWD3 each perform worst four times overall.

### 3.1.4 Sensitivity to Ensemble Size

So far, we used all ensemble members at once to fit our candidate PDFs onto simulated precipitation. That improves the quality of the fit. In this section, we first analyze a single ensemble member and investigate subsequently the sensitivity of our candidate PDFs' performance on the ensemble size. In doing so, we properly disentangle the difference between observations and simulations from the impact of the sample size.

As before, 3-parameter candidate distribution functions also perform for a single ensemble simulation better than 2-parameter PDFs (Table 4). For a single ensemble member, the difference by which 3-parameter PDFs out-perform 2-parameter PDFs reduces considerably relative to the entire ensemble simulations (compare Table 4 against Table 3), though. In contrast to Table 3, all of our candidate distribution functions perform similarly between a single ensemble simulation and observations. In contrast to our previous results (e.g. when analyzing weighted sums of deviations from $\mathcal{N}_{0,1}$), modeled and observed precipitation distributions now seem sufficiently similar. Reducing the sample size for the fit by a factor of ten leads to more homogeneous performances of all candidate PDFs in simulations. As a reminder, AIC-D frequencies as depicted in Table 4 measure only relative performance differences. Consequently, our 2-parameter candidate PDFs do not actually perform better with fewer data. Instead, limiting the input data to a single ensemble member impairs our 3-parameter candidate PDFs stronger than our 2-parameter candidate PDFs. Optimizing 3 parameters needs more information than the optimization of 2 parameters. Irrespective of the realization, GD2 performs erroneously for 31 samples (apparent in grid-points which display AIC-D values beyond

4). Despite the need for more information, 31 samples suffice EWD3 to fix GD2's erroneous performances in both analyzed realizations.

In the next step, we isolate and investigate the improvement of the fit by an increasing sample/ensemble size. As a consequence of limited observed global precipitation data, we neglect observations and their differences to simulations in this remaining section. During this investigation, we reanalyze Table 4 while iteratively increasing the ensemble (sample) size for the fit (and the AIC-D calculation). Irrespective of the ensemble size, EWD3 performs proficiently (Table 5). Further, the fraction of grid-points in which EWD3 performs ideal increases constantly. This is a consequence of EWD3's better performance

relative to our 2-parameter candidate PDFs. Unfortunately, AIC-Ds can only compare models that are based on an equal sample size without adhering to additional undesired assumptions. Thus, any direct analysis of each candidate PDF's improvement relative to its own performance for a single ensemble member is with AIC-D frequencies not feasible. Despite this caveat, Table 5 still indicates strongly that EWD3 benefits stronger from the increased sample size than any of our 2-parameter candidate distribution functions. The larger the sample size, the larger is the margin by which EWD3 outperforms GD2.

Despite requiring more data, our 3-parameter candidate PDFs perform already better for 31 samples. For 31 samples, we identify this better performance of 3-parameter candidate PDFs in observations and simulations. Further, since our 3-parameter candidate PDFs require more data to estimate optimal parameters, they benefit in simulations stronger from additional samples than our 2-parameter candidate PDFs. That benefit becomes apparent in a distinctly improved relative performance after multiplying the sample size through the use of additional ensemble members.

## 3.2    Other SPI Accumulation Periods

A similar pattern as identified for $SPI_{3M}$ also emerges in the evaluation of AIC-D-based performances of our candidate PDFs for accumulation periods of 1-, 6-, 9-, and 12-months (Table 7). No candidate PDF performs ideally (AIC-D values below 2) with substantial confidence around the globe. The reasons for this shortcoming are distribution-dependent. GD2 performs too poor in too many grid-points (e.g. apparent by too low percentages for covering AIC-D values even below 4) and EWD3

excels only for AIC-D values beyond 2 because it first needs to over-compensate its AIC-imposed complexity-penalty (as explained before). Equally apparent is the striking inability of the 2-parameter candidate PDFs to adequately perform in ensemble simulations for all analyzed accumulation periods which we have also seen for $SPI_{3M}$ before.

     In agreement with prior studies (Stagge et al., 2015; Sienz et al., 2012), we also identify the apparent performance shift between short (less than 3-months) and long (more than 3-months) accumulation periods for the 2-parameter candidate PDFs.

While WD2 performs well for short accumulation periods (only in observations though), GD2 performs better than WD2 for longer accumulation periods. Nevertheless, neither 3-parameter candidate PDF displays such a shift in its performance. Both 3-parameter PDFs perform for accumulation periods shorter and longer than 3-months similarly well.

     Most interesting, EWD3 performs well almost everywhere around the entire globe for each accumulation period and in both realizations. EWD3 shows the highest percentages of all candidate PDFs for each analysis (each row of Table 6) beyond AIC-D

values of 2; except for an accumulation period of 12-months in simulations. While there is not even a single candidate PDF that seems well suited for an accumulation period of 12-months in simulations, GD2 and EWD3 both perform equally adequate;

despite EWD3's higher AIC-penalty compared to GD2. As a reminder, AIC punishes EWD3 stronger than GD2. Nevertheless this complexity punishment, it is obvious by now that our 2-parameter PDFs are inept to universally deliver normal distributed SPI time-series; particularly if one considers all depicted dimensions of the task at hand. As it turns out, this punishment is

the sole reason for both performance limitations that EWD3 displays in Table 6: (i) for the ideal AIC-D category and (ii) EWD3's tied performance with GD2 for an accumulation period of 12-months in ensemble simulations. As shown before, AIC's punishment is particularly noticeable in the ideal category. Further, this punishment also affects the tied performance ranking for the accumulation period of 12-months. To illustrate this effect, we again consider AIC's estimated likelihood (without its penalty) to correct EWD3's performance for the complexity punishment. While we again analytically analyzed this

consideration, for the scope of this publication a first-order approximation suffices also here. In that first-order approximation of this consideration, EWD3's coverages of Table 6 shift again by 2.46 (2.04) AIC units in observations (ensemble simulations). Since neighboring AIC-D categories differ by 2-3 AIC units, this approximation shifts EWD3's coverages of Table 6 by roughly one category. Such a shift would solve EWD3's limitation in the ideal AIC-D category. Further, EWD3 would also perform best across all AIC-D categories in ensemble simulations; including the accumulation period of 12-months.

Despite the inclusion of the complexity penalty, EWD3 performs still best in 32 out of all 40 analyses (all rows of Table 3 and Table 6), and in 30 of those 32 analyses, we consider EWD3's performance to display at least average confidence (indicated by a yellow or green background color in the respective table). In contrast, GD2 only performs 7 (2) times best (while also performing with at least average confidence) – WD2 performs once best and GGD3 never.

## 4   Discussion

Previous studies have emphasized the importance of using a single PDF to calculate SPI for each accumulation period and location (Stagge et al., 2015; Guttman, 1999) to ensure comparability across space and time which is one of the index's main advantages (Lloyd-Hughes and Saunders, 2002). However, any 2-parameter distribution function seems in observations already ill-suited to deliver adequately normally distributed SPI time-series. Single 2-parameter candidate PDFs are neither suited for all locations nor both short (less than 3-months) and long (more than 3-months) accumulation periods (Stagge

et al., 2015; Sienz et al., 2012). Introducing ensemble simulations as another level of complexity exacerbates the problem additionally. Yet, the importance of accepting and solving this problem becomes increasingly pressing as a result of a growing interest in dynamical drought predictions and their evaluation against observations. To properly evaluate drought predictability of precipitation hindcasts against observations, the distribution function used in SPI's calculation algorithm needs to capture sufficiently well both frequency distributions mutually: those of observed and modeled precipitation totals.

The outlined problem is additionally aggravated by the fact that it cannot be circumnavigated. Our results demonstrate that any inept description of precipitation by SPI's candidate distribution function manifests most severely in the tails of SPI's distribution. Since SPI is usually employed to analyze the left-hand tail of its distribution (droughts), biased descriptions of this tail are highly undesirable. To establish the robustness of this valuable tool and to fully capitalize its advantages, SPI's problem of requiring a single, universally applicable candidate PDF needs to be solved. In this study, we show that the 3-parameter

exponentiated Weibull distribution (EWD3) is very promising in solving this problem virtually everywhere around the globe in both realizations (observations and simulations) for all common accumulation periods (1-, 3-, 6-, 9-, and 12-months).

Other studies have dismissed the possibility of such a solution to this problem and proposed instead a multi-PDF approach (Guenang et al., 2019; Blain and Meschiatti, 2015; Touma et al., 2015; Sienz et al., 2012; Lloyd-Hughes and Saunders, 2002) which selects different PDFs depending on the location and accumulation period of interest. The emergence of this proposal

stems from a focus on 2-parameter PDFs that exhibit a shift in their performance which depends on the scrutinized accumulation period. While WD2 performs better for an accumulation period of 1-month, GD2 is better suited for longer accumulation periods. However, any multi-PDF approach would partly sacrifice the aforementioned index's pivotal advantage of comparability across space and time. Our results suggest that such a multi-PDF approach does not improve the normality of calculated SPI time-series relative to a calculation algorithm that uses EWD3 as PDF everywhere. Furthermore, the use of an empiri-

cal cumulative distribution function has been proposed (Sienz et al., 2012). We checked this approach which proved to be too coarse because of its discretized nature (not shown). As a result of its discretized nature, the analyzed sample size prescribes the magnitude of deviations from $\mathcal{N}_{0,1}$. Consequently, these deviations are spatially invariant and aggregate with each additional grid-point. Thus, deviations from $\mathcal{N}_{0,1}$ will not spatially balance each other.

Yet, in agreement with those other studies (Guenang et al., 2019; Blain and Meschiatti, 2015; Touma et al., 2015; Sienz

et al., 2012; Lloyd-Hughes and Saunders, 2002), our results also suggest that 2-parameter PDFs are inept for all accumulation periods, locations, and realizations. Despite this inability of 2-parameter PDFs, EWD3 competed against 2-parameter PDFs in our analysis. This competition unnecessarily (given the inadequacy of 2-parameter PDFs, the risk of underfitting seems to outweigh the risk of overfitting) exacerbates EWD3's performance assessed with AIC-D because AIC punishes complexity (irrespective of that risk consideration). As a consequence of EWD3's increased complexity, AIC imposes a larger penalty on

EWD3 than on the 2-parameter candidate PDFs (which are anyhow ill-suited to solve the outlined problem because they are most likely too simple). Still, EWD3 conclusively out-performs any other candidate PDF. Yet, EWD3 does not perform ideally with substantial confidence in ensemble simulations. However, leveling the playing field for candidate distribution functions with different parameter counts in our AIC-D analysis leads to an ideal performance of EWD3 universally.

We also repeated our AIC-D analysis with the Bayesian information criterion (Schwarz et al., 1978) which delivered similar

results. Irrespective of the employed information criterion, the findings sketched above stay valid on every continent in both realizations with a few exceptions. It seems noteworthy, that Australia's observed DJF- and modeled JJA-precipitation totals are generally poorly described by any of our candidate distribution functions. Since the performances of all investigated distribution functions deteriorate to a similar level, it is difficult, however, to discern any new ranking. Even more troublesome is the proper description of simulated 12-months precipitation totals. Here, our candidate PDFs perform only sufficiently. Yet, despite its

increased AIC-penalty, EWD3 performs still best along with the 2-parameter gamma distribution.

Overall our 3-parameter candidate PDFs perform better than investigated 2-parameter candidate PDFs. Despite requiring more data, a sample size of 31 years suffices our 3-parameter candidate PDFs to outperform our 2-parameter candidate PDFs in simulations and observations. Further, our 3-parameter candidate PDFs greatly benefit from an increase in the sample size in simulations. In simulations, such a sample size sensitivity analysis is feasible by exploiting different counts of ensemble

members. Whether 3-parameter PDFs would benefit similarly from an increased sample size in observations is likely but ultimately remains speculative because trustworthy global observations of precipitation are temporally too constrained for such a sensitivity analysis.

In contrast to Blain et al. (2018), who investigated the influence of different parameter estimation methods on the normality of the resulting SPI time-series and only found minuscule effects, our results show a substantial impact of the meticulousness applied to optimize the same parameter estimation method. Despite using the same parameter estimation methods and the same candidate PDF, the baseline investigated here enlarges deviations from $\mathcal{N}_{0,1}$ by roughly half a magnitude compared to GD2 in DJF. This result is concerning because it indicates that main differences do not only emerge when using different parameter estimation methods but rather manifest already in the applied procedures by which these methods are optimized. In our analysis, not different PDFs but different optimization procedures of the same parameter estimation method can impact normality most profoundly.

Other consequences of this finding are apparent major season-dependent differences in the performance of the investigated baseline. This finding contradicts the results of Stagge et al. (2015) (and the results we obtained from the analysis of our candidate PDFs). These results suggest that the performances of candidate PDFs are independent of the season. In contrast, the baseline performs similar to GD2 during JJA, but the performance of the baseline severely deteriorates during DJF in our analysis. While this deterioration is overall more apparent in observations than in simulations, its most obvious instance occurs in simulations. The investigated baseline over-estimates modeled extreme droughts in Australia during DJF by more than 240% – that depicts the largest deviation from $\mathcal{N}_{0,1}$ we encountered in this study. Therefore, we urge to exercise substantial caution while analyzing $\text{SPI}_{DJF}$ time-series with the investigated baseline's R-package irrespective of the heritage of input data. While the largest deviations from $\mathcal{N}_{0,1}$ occur during DJF in Australia, the baseline performs particularly poorly during DJF in general. During DJF, the examined baseline displays larger deviations from $\mathcal{N}_{0,1}$ than any other of the six here analyzed SPI calculations (GD2, WD2, GGD3, EWD3, baseline, and $\text{AIC}_{min}$-analysis) in 63 out of 98 different analyses, which range across all seven SPI categories, all seven regions, and both realizations. Aside from the investigated baseline and in general agreement with Stagge et al. (2015), we find only in Australia minor seasonal differences in the performance of our candidate PDFs.

To aggregate our AIC-D-analysis over the globe and visualize this aggregation in tables, we need to evaluate the aggregated performance of candidate PDFs for certain AIC-D categories (Burnham and Anderson, 2002). Their aggregation over all land grid-points of the globe demands the introduction of another performance criterion that requires interpretation. That criterion informs whether the candidate PDFs conform to the respective AIC-D categories in sufficient grid-points globally and, therefore, needs to interpret which fraction of the global land grid-points can be considered sufficient. For this fraction of global land grid-points, we select 85 % and 95 % as thresholds. Consequently, we categorize our candidate PDFs for each AIC-D category into three different classes of possible performances. We consider the confirmation of the respective AIC-D category in at least 95% of grid-points globally as an indicator of substantial confidence in the candidate PDF to perform according to the respective AIC-D category globally. Confirmation of the respective AIC-D category in less than 85% of grid-points globally is considered as an indicator of insufficient confidence in the candidate PDF. Lastly, we consider it to be an

indicator of average confidence in candidate PDFs when they conform to the respective AIC-D category in between 85% and 95% of grid-points globally. One might criticize that these thresholds lack a scientific foundation or that they are to some extent arbitrary. However, they seem adequately reasonable and agree with analog evaluations of such fractions derived by rejection frequencies from goodness-of-fit tests in previous studies (Blain et al., 2018; Blain and Meschiatti, 2015; Stagge et al., 2015; Lloyd-Hughes and Saunders, 2002). Moreover, these thresholds show a robust statistical basis in terms of being equally represented over all 320 analyzed evaluations in this study (all entries of Table 3, Table 4. Table 5, and Table 6). Across all 80 analyses (all rows of Table 3, Table 4, Table 5, and Table 6), the four candidate PDFs perform insufficiently 132 times, while they perform with substantial (average) confidence 130 (58) times.

There is scope to further test the robustness of our derived conclusions in different models with different time horizons and foci on accumulation periods other than 3-months (e.g. 12-months). Of additional interest would be insights about the distribution of precipitation. Such insights would enable SPI's calculation algorithm to physically base its key decision. A recent study suggests that a 4-parameter extended generalized Pareto distribution excels in describing the frequency distribution of precipitation (Tencaliec et al., 2020). Anyhow, the inclusion of yet another distribution parameter additionally complicates the optimization of the parameter estimation method. We already exemplified the impact of the meticulousness of the applied optimization in this study. Establishing a standard for the optimization process seems currently more urgent than attempts to improve SPI through 4-parameter PDFs.

The results presented here further imply that the evaluated predictive skill of drought predictions assessed with SPI should be treated with caution because it is likely biased by SPI's current calculation algorithms. This common bias in SPI's calculation algorithms obscures the evaluation of predictive skill of ensemble simulations by inducing a blurred representation of their precipitation distributions. That blurred representation emerges in the simulated drought index which impedes the evaluation process. Drought predictions often try to correctly predict the drought intensity. The evaluation process usually considers this to be successfully achieved if the same SPI category as the observed one is predicted. This evaluation is quite sensitive to the thresholds used when classifying SPI categories. The bias identified here blurs these categories in ensemble simulations stronger than in observations against which the model's predictability is customarily evaluated. As a consequence of these sensitive thresholds, such a one-sided bias potentially undermines current evaluation processes.

## 5   Summary and Conclusions

Current SPI calculation algorithms are tailored to describe observed precipitation distributions. Consequently, current SPI calculation algorithms are ineptly suited to describe precipitation distributions obtained from ensemble simulations. Also in observations, erroneous performances are apparent and well-known, but less conspicuous than in ensemble simulations. We propose a solution that rectifies these issues and improves the description of modeled and observed precipitation distributions individually as well as concurrently. The performance of 2-parameter candidate distribution functions is inadequate for this task. By increasing the parameter count of the candidate distribution function (and thereby also its complexity) a distinctly better description of precipitation distributions can be achieved. In simulations and observation, the here identified best-performing

candidate distribution function – the exponentiated Weibull distribution (EWD3) – performs proficiently for every common accumulation period (1-, 3-, 6-, 9-, and 12-months) virtually everywhere around the globe. Additionally, EWD3 excels when analyzing ensemble simulations. Its increased complexity (relative to GD2) leads to an outstanding performance of EWD3 when an available ensemble multiplies the sample size.

We investigate different candidate distribution functions (gamma (GD2), Weibull (WD2), generalized gamma (GGD3), and exponentiated Weibull distribution (EWD3)) in SPI's calculation algorithm and evaluate their adequacy in meeting SPI's normality requirement. We conduct this investigation for observations and simulations during summer (JJA) and winter (DJF). Our analysis evaluates globally and over each continent individually the resulting $SPI_{3M}$ time-series based on their normality. This analysis focuses on an accumulation period of 3-months and tests the conclusions drawn from that focus for the most common other accumulation periods (1-, 6-, 9-, and 12-months). The normality of SPI is assessed by two complementary analyses. The first analysis checks the absolute performance of candidate PDFs by comparing actual occurrence probabilities of SPI categories (as defined by WMO's *SPI User Guide* (Svoboda et al., 2012)) against well-known theoretical occurrence probabilities of $\mathcal{N}_{0,1}$. The second analysis evaluates candidate PDFs relative to each other while penalizing unnecessary complexity with Akaike's Information Criterion (AIC).

Irrespective of the accumulation period or the data-set, GD2 seems sufficiently suited to be employed in SPI's calculation algorithm in many grid-points of the globe. Yet, GD2 also performs erroneous in a non-negligible fraction of grid-points. These erroneous performances are apparent in observations and simulations for each accumulation period. More severely, GD2's erroneous performances decline further in ensemble simulations. Here, GD2 performs in a non-negligible fraction of grid-points also insufficient or even without any skill. In contrast, EWD3 performs for all accumulation periods without any defects, irrespective of the data-set. Despite requiring more data than 2-parameter PDFs, we identify EWD3's proficient performance for a sample size of 31 years in observations as well as in simulations. Further, ensemble simulations allow us to artificially increase the sample size for the fitting procedure by including additional ensemble members. Exploiting this possibility has a major impact on the performance of candidate PDFs. The margin, by which EWD3 outperforms GD2, further increases with additional ensemble members. Furthermore, EWD3 demonstrates proficiency also for every analyzed accumulation period around the globe. The accumulation period of 12-months poses in simulations the only exception. Here, EWD3 and GD2 both perform similarly well around the globe. Still, we find that 3-parameter PDFs are generally better suited in SPI's calculation algorithm than 2-parameter PDFs.

Given all the dimensions (locations, realizations, accumulation periods) of the task, our results suggest that the risk of underfitting by using 2-parameter PDFs is larger than the risk of overfitting by employing 3-parameter PDFs. We strongly advocate adapting the calculation algorithm of SPI and the therein use of 2-parameter distribution functions in favor of 3-parameter PDFs. Such an adaptation is particularly important for the proper evaluation and interpretation of drought predictions derived from ensemble simulations. For this adaptation, we propose the employment of EWD3 as a new standard PDF for SPI's calculation algorithm, irrespective of the heritage of input data or the length of scrutinized accumulation periods. Despite the issues discussed here, SPI remains a valuable tool for analyzing droughts. This study might contribute to the value of this tool by illuminating and resolving the discussed long-standing issue concerning the proper calculation of the index.

*Data availability.* The model simulations are available at the World Data Center for Climate (WDCC): http://cera-www.dkrz.de/WDCC/ui/Compact.jsp?acronym=DKRZ_LTA_1075_ds00001 maintained by the Deutsche Klimarechenzentrum (DKRZ, German Climate Computing Centre).

830

*Author contributions.* PP, AD, and JB designed the study. PP led the analysis and prepared the manuscript with support from all co-authors. All co-authors contributed to the discussion of the results.

*Competing interests.* The authors declare that they have no conflict of interest.

*Acknowledgements.* This work was funded by the BMBF-funded joint research projects RACE – Regional Atlantic Circulation and Global Change and RACE – Synthesis. P.P. is supported by the Stiftung der deutschen Wirtschaft (SDW, German Economy Foundation). A.D. and J.B. are supported by the Deutsche Forschungsgemeinschaft (DFG, German Research Foundation) under Germany's Excellence Strategy– EXC 2037 "Climate, Climatic Change, and Society"–Project: 390683824, contribution to the Center for Earth System Research and Sustainability (CEN) of Universität Hamburg. A.D. is also supported by A4 (Aigéin, Aeráid, agus athrú Atlantaigh), funded by the Marine Institute and the European Regional Development fund (grant: PBA/CC/18/01). The model simulations were performed at the German Climate Computing Centre. The authors also thank Frank Sienz for providing the software to compute AIC and SPI with different candidate distribution functions. The authors would also like to thank Gabriel Blain and another anonymous referee for their effort in reviewing this paper and editor Marie-Claire ten Veldhuis for her engagement in overseeing and actively participating in the review process.

835

840

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

**Table 1.** Abbreviations used for candidate distribution functions.

| Distribution function | Parameter count | Abbreviation |
|---|---|---|
| Gamma distribution | 2 | GD2 |
| Weibull distribution | 2 | WD2 |
| Generalized gamma distribution | 3 | GGD3 |
| Exponentiated Weibull distribution | 3 | EWD3 |

**Table 2.** Standardized Precipitation Index (SPI) classes with their corresponding SPI intervals and theoretical occurrence probabilities (according to WMO's *SPI User Guide* (Svoboda et al., 2012)).

| SPI interval | SPI class | Probability [%] |
|---|---|---|
| $SPI \geq 2$ | W3: extremely wet | 2.3 |
| $2 > SPI \geq 1.5$ | W2: severely wet | 4.4 |
| $1.5 > SPI \geq 1$ | W1: moderately wet | 9.2 |
| $1 > SPI > -1$ | N0: normal | 68.2 |
| $-1 \geq SPI > -1.5$ | D1: moderately dry | 9.2 |
| $-1.5 \geq SPI > -2$ | D2: severely dry | 4.4 |
| $SPI \leq -2$ | D3: extremely dry | 2.3 |

**Table 3.** Percent of grid-points that are classified into specific AIC-D categories (according to Burnham and Anderson (2002)) for each candidate PDF over both seasons. Percentages of grid-points indicate the confidence in candidate PDFs to overall perform according to the respective AIC-D category. We consider percentages that exceed (subceed in case of AIC-D values beyond 10) 95% (5%) as a sign of substantial confidence in the candidate PDF (green) to overall perform according to the respective AIC-D category. In contrast, we consider those candidate PDFs that exceed/subceed in 85/15% of the grid-points as a sign of average confidence in the candidate PDF (yellow) to overall perform according to the respective AIC-D category. Percentages that fall short of 85% (or that show no skill in more than 15%) are considered as an overall sign of insufficient confidence in the candidate PDF (red).

| SPI Period | Realization | AIC-D category | GD2 | WD2 | GGD3 | EWD3 |
|---|---|---|---|---|---|---|
| 3-Months | Observations | Ideal (AIC-D $\leq$ 2) | 84 | 76 | 22 | 31 |
| | | Well (AIC-D $\leq$ 4) | 94 | 91 | 98 | 100 |
| | | Sufficient (AIC-D $\leq$ 7) | 98 | 98 | 100 | 100 |
| | | No Skill (AIC-D > 10) | 1 | 0 | 0 | 0 |
| | Ensemble Simulations | Ideal (AIC-D $\leq$ 2) | 65 | 18 | 68 | 86 |
| | | Well (AIC-D $\leq$ 4) | 74 | 24 | 89 | 99 |
| | | Sufficient (AIC-D $\leq$ 7) | 82 | 34 | 94 | 99 |
| | | No Skill (AIC-D > 10) | 12 | 57 | 4 | 1 |

**Table 4.** Percent of grid-points that are classified into specific AIC-D categories (according to Burnham and Anderson (2002)) for each candidate PDF over both seasons. Percentages of grid-points indicate the confidence in candidate PDFs to overall perform according to the respective AIC-D category. We consider percentages that exceed (subceed in case of AIC-D values beyond 10) 95% (5%) as a sign of substantial confidence in the candidate PDF (green) to overall perform according to the respective AIC-D category. In contrast, we consider those candidate PDFs that exceed/subceed in 85/15% of the grid-points as a sign of average confidence in the candidate PDF (yellow) to overall perform according to the respective AIC-D category. Percentages that fall short of 85% (or that show no skill in more than 15%) are considered as an overall sign of insufficient confidence in the candidate PDF (red). In contrast to Table 3, the evaluation of simulations is based on a single ensemble member. Observations are identical to Table 3.

| SPI Period | Realization | AIC-D category | GD2 | WD2 | GGD3 | EWD3 |
|---|---|---|---|---|---|---|
| 3-Months | Observations | Ideal (AIC-D $\leq 2$) | 84 | 76 | 22 | 31 |
| | | Well (AIC-D $\leq 4$) | 94 | 91 | 98 | 100 |
| | | Sufficient (AIC-D $\leq 7$) | 98 | 98 | 100 | 100 |
| | | No Skill (AIC-D $> 10$) | 1 | 0 | 0 | 0 |
| | Single Ensemble Member | Ideal (AIC-D $\leq 2$) | 83 | 76 | 19 | 28 |
| | | Well (AIC-D $\leq 4$) | 93 | 92 | 98 | 100 |
| | | Sufficient (AIC-D $\leq 7$) | 98 | 98 | 100 | 100 |
| | | No Skill (AIC-D $> 10$) | 1 | 0 | 0 | 0 |

**Table 5.** Percent of grid-points that are classified into specific AIC-D categories (according to Burnham and Anderson (2002)) for each candidate PDF over both seasons. Percentages of grid-points indicate the confidence in candidate PDFs to overall perform according to the respective AIC-D category. We consider percentages that exceed (subceed in case of AIC-D values beyond 10) 95% (5%) as a sign of substantial confidence in the candidate PDF (green) to overall perform according to the respective AIC-D category. In contrast, we consider those candidate PDFs that exceed/subceed in 85/15% of the grid-points as a sign of average confidence in the candidate PDF (yellow) to overall perform according to the respective AIC-D category. Percentages that fall short of 85% (or that show no skill in more than 15%) are considered as an overall sign of insufficient confidence in the candidate PDF (red). In contrast to Table 3, the evaluation of simulations is based on different ensemble sizes.

| SPI Period | Ensemble Size | AIC-D category | GD2 | WD2 | GGD3 | EWD3 |
|---|---|---|---|---|---|---|
| | 2 | Ideal (AIC-D ≤ 2) | 78 | 56 | 43 | 57 |
| | | Well (AIC-D ≤ 4) | 87 | 74 | 96 | 99 |
| | | Sufficient (AIC-D ≤ 7) | 94 | 90 | 98 | 100 |
| | | No Skill (AIC-D > 10) | 3 | 4 | 1 | 0 |
| | 3 | Ideal (AIC-D ≤ 2) | 77 | 45 | 53 | 69 |
| | | Well (AIC-D ≤ 4) | 86 | 61 | 96 | 99 |
| | | Sufficient (AIC-D ≤ 7) | 93 | 79 | 99 | 100 |
| | | No Skill (AIC-D > 10) | 4 | 10 | 1 | 0 |
| | 4 | Ideal (AIC-D ≤ 2) | 75 | 38 | 59 | 74 |
| | | Well (AIC-D ≤ 4) | 84 | 50 | 95 | 99 |
| | | Sufficient (AIC-D ≤ 7) | 90 | 67 | 98 | 100 |
| | | No Skill (AIC-D > 10) | 7 | 19 | 2 | 0 |
| | 5 | Ideal (AIC-D ≤ 2) | 74 | 31 | 63 | 79 |
| | | Well (AIC-D ≤ 4) | 82 | 42 | 94 | 99 |
| | | Sufficient (AIC-D ≤ 7) | 89 | 57 | 97 | 99 |
| | | No Skill (AIC-D > 10) | 7 | 30 | 2 | 0 |
| 3-Months | 6 | Ideal (AIC-D ≤ 2) | 73 | 27 | 64 | 80 |
| | | Well (AIC-D ≤ 4) | 81 | 36 | 93 | 99 |
| | | Sufficient (AIC-D ≤ 7) | 88 | 50 | 96 | 99 |
| | | No Skill (AIC-D > 10) | 9 | 37 | 2 | 0 |
| | 7 | Ideal (AIC-D ≤ 2) | 70 | 25 | 66 | 81 |
| | | Well (AIC-D ≤ 4) | 78 | 33 | 92 | 98 |
| | | Sufficient (AIC-D ≤ 7) | 86 | 45 | 96 | 99 |
| | | No Skill (AIC-D > 10) | 10 | 43 | 2 | 1 |
| | 8 | Ideal (AIC-D ≤ 2) | 69 | 21 | 67 | 83 |
| | | Well (AIC-D ≤ 4) | 77 | 29 | 91 | 98 |
| | | Sufficient (AIC-D ≤ 7) | 85 | 39 | 95 | 99 |
| | | No Skill (AIC-D > 10) | 11 | 49 | 3 | 1 |
| | 9 | Ideal (AIC-D ≤ 2) | 66 | 20 | 67 | 85 |
| | | Well (AIC-D ≤ 4) | 76 | 27 | 90 | 99 |
| | | Sufficient (AIC-D ≤ 7) | 84 | 36 | 95 | 99 |
| | | No Skill (AIC-D > 10) | 12 | 53 | 3 | 1 |

**Table 6.** Percent of grid-points that are classified into specific AIC-D categories (according to Burnham and Anderson (2002)) for each candidate PDF over both seasons. Percentages of grid-points indicate the confidence in candidate PDFs to overall perform according to the respective AIC-D category. We consider percentages that exceed (subceed in case of AIC-D values beyond 10) 95% (5%) as a sign of substantial confidence in the candidate PDF (green) to overall perform according to the respective AIC-D category. In contrast, we consider those candidate PDFs that exceed/subceed in 85/15% of the grid-points as a sign of average confidence in the candidate PDF (yellow) to overall perform according to the respective AIC-D category. Percentages that fall short of 85% (or that show no skill in more than 15%) are considered as an overall sign of insufficient confidence in the candidate PDF (red). In contrast to Table 3, this table evaluates different accumulations periods of SPI.

| SPI Period | Realization | AIC-D category | GD2 | WD2 | GGD3 | EWD3 |
|---|---|---|---|---|---|---|
| 1-Month | Observations | Ideal (AIC-D ≤ 2) | 84 | 86 | 30 | 33 |
| | | Well (AIC-D ≤ 4) | 94 | 97 | 100 | 100 |
| | | Sufficient (AIC-D ≤ 7) | 98 | 99 | 100 | 100 |
| | | No Skill (AIC-D > 10) | 0 | 0 | 0 | 0 |
| | Ensemble Simulations | Ideal (AIC-D ≤ 2) | 55 | 43 | 81 | 87 |
| | | Well (AIC-D ≤ 4) | 64 | 54 | 96 | 100 |
| | | Sufficient (AIC-D ≤ 7) | 73 | 66 | 98 | 100 |
| | | No Skill (AIC-D > 10) | 21 | 26 | 1 | 0 |
| 6-Months | Observations | Ideal (AIC-D ≤ 2) | 82 | 67 | 16 | 30 |
| | | Well (AIC-D ≤ 4) | 93 | 86 | 96 | 99 |
| | | Sufficient (AIC-D ≤ 7) | 99 | 98 | 99 | 100 |
| | | No Skill (AIC-D > 10) | 0 | 0 | 0 | 0 |
| | Ensemble Simulations | Ideal (AIC-D ≤ 2) | 75 | 11 | 49 | 77 |
| | | Well (AIC-D ≤ 4) | 82 | 15 | 82 | 95 |
| | | Sufficient (AIC-D ≤ 7) | 88 | 22 | 90 | 97 |
| | | No Skill (AIC-D > 10) | 8 | 71 | 7 | 2 |
| 9-Months | Observations | Ideal (AIC-D ≤ 2) | 83 | 64 | 13 | 28 |
| | | Well (AIC-D ≤ 4) | 93 | 84 | 93 | 98 |
| | | Sufficient (AIC-D ≤ 7) | 99 | 97 | 98 | 99 |
| | | No Skill (AIC-D > 10) | 0 | 1 | 1 | 0 |
| | Ensemble Simulations | Ideal (AIC-D ≤ 2) | 75 | 10 | 40 | 76 |
| | | Well (AIC-D ≤ 4) | 82 | 13 | 76 | 93 |
| | | Sufficient (AIC-D ≤ 7) | 89 | 18 | 85 | 95 |
| | | No Skill (AIC-D > 10) | 7 | 76 | 12 | 3 |
| 12-Month | Observations | Ideal (AIC-D ≤ 2) | 82 | 61 | 13 | 29 |
| | | Well (AIC-D ≤ 4) | 92 | 81 | 91 | 96 |
| | | Sufficient (AIC-D ≤ 7) | 98 | 96 | 97 | 98 |
| | | No Skill (AIC-D > 10) | 1 | 1 | 1 | 1 |
| | Ensemble Simulations | Ideal (AIC-D ≤ 2) | 79 | 9 | 34 | 69 |
| | | Well (AIC-D ≤ 4) | 86 | 11 | 75 | 87 |
| | | Sufficient (AIC-D ≤ 7) | 91 | 15 | 83 | 90 |
| | | No Skill (AIC-D > 10) | 6 | 80 | 14 | 7 |

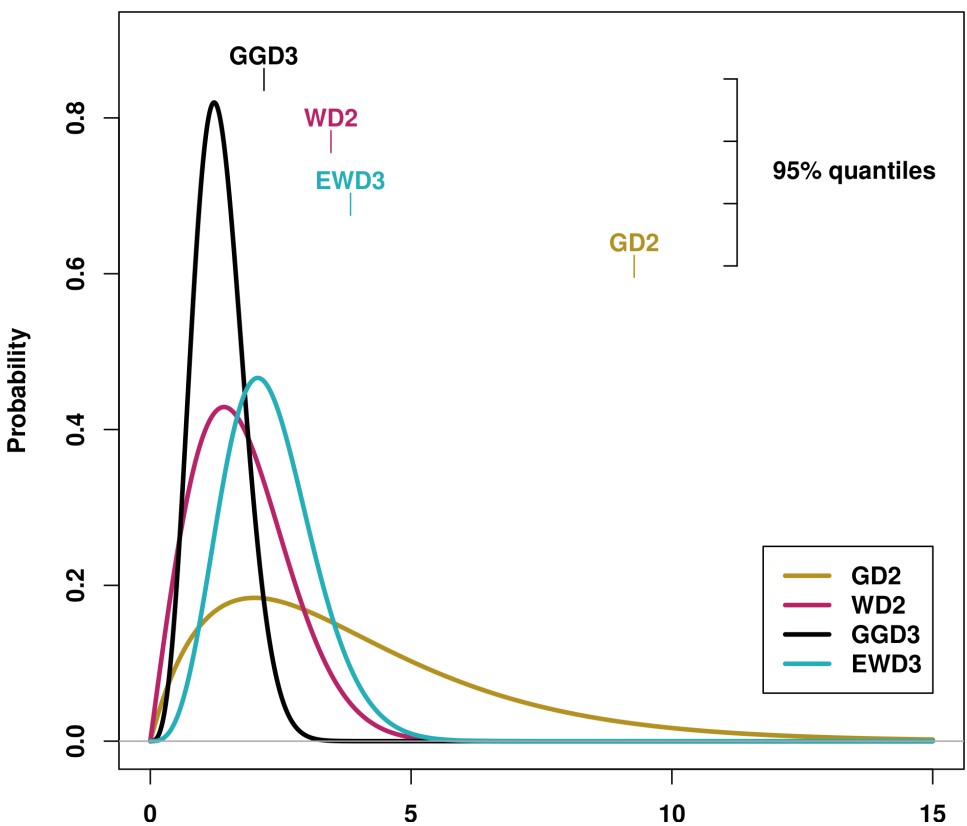

**Figure 1.** Candidate Distribution functions whose performance is investigated in this study: the 2-parameter gamma distribution (GD2), the 2-parameter Weibull distribution (WD2), the 3-parameter generalized gamma distribution (GGD3) and the 3-parameter exponentiated Weibull distribution (EWD3). Displayed are examples of those PDFs for $\sigma = \gamma (= \alpha) = 2$ and their corresponding 95% quantiles.

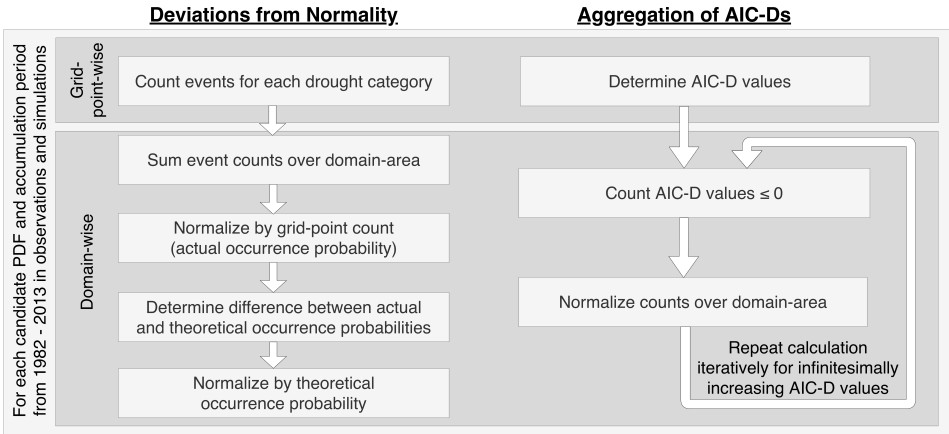

**Figure 2.** Flow chart of methods to aggregate deviations from $\mathcal{N}_{0,1}$ (**left**) and AIC-D frequencies (**right**) over domains.

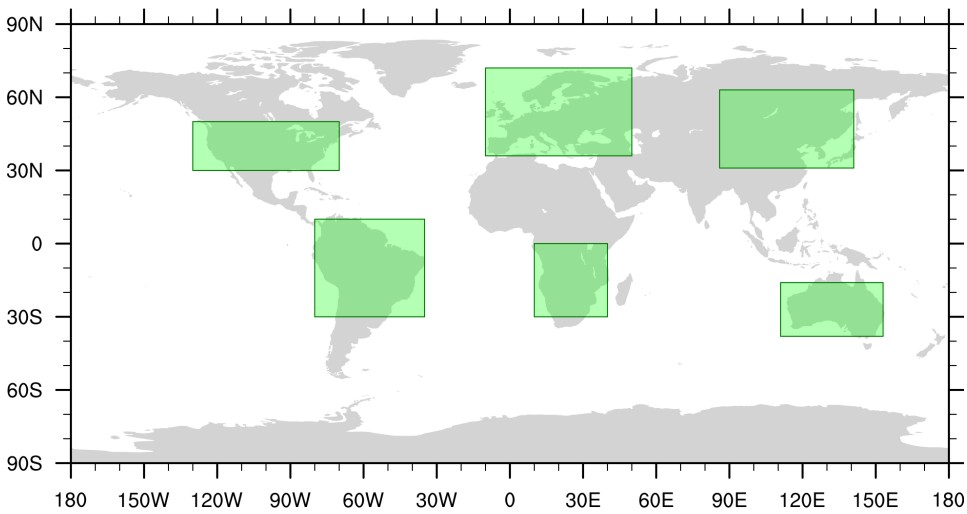

**Figure 3.** Borders of regions examined in this study.

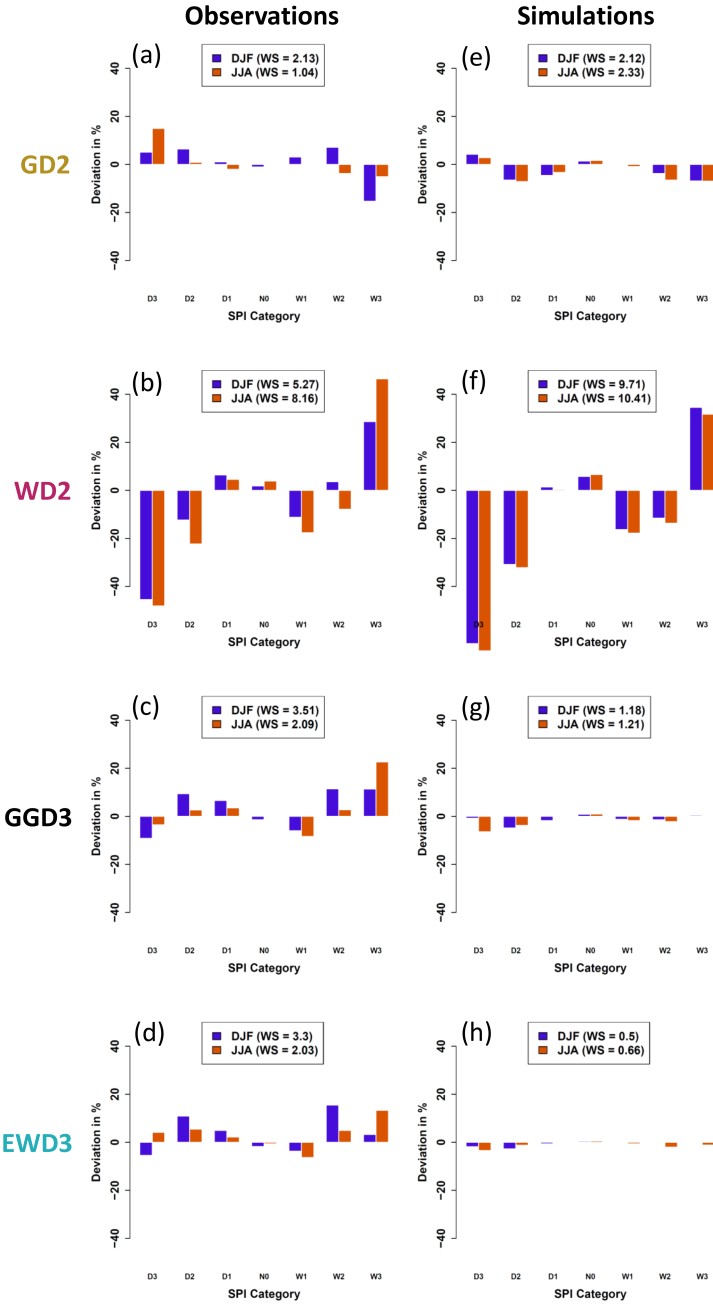

**Figure 4.** Deviations from $\mathcal{N}_{0,1}$ over the entire globe for observed (**left**) and modeled (**right**) SPI time-series. SPI time-series are derived by using the simple 2-parameter gamma distribution (GD2, **top row**), the simple 2-parameter Weibull distribution (WD2, **second row**), the 3-parameter generalized gamma distribution (GGD3, **third row**), and the 3-parameter exponentiated Weibull distribution (EWD3, **bottom row**). The legends depict weighted (by their respective theoretical occurrence probability) sums (WS) of deviations from $\mathcal{N}_{0,1}$ over all SPI categories. Irrespective of the candidate PDF, deviations from $\mathcal{N}_{0,1}$ are smallest for the center of SPI's distribution (N0) and largest for its tails.

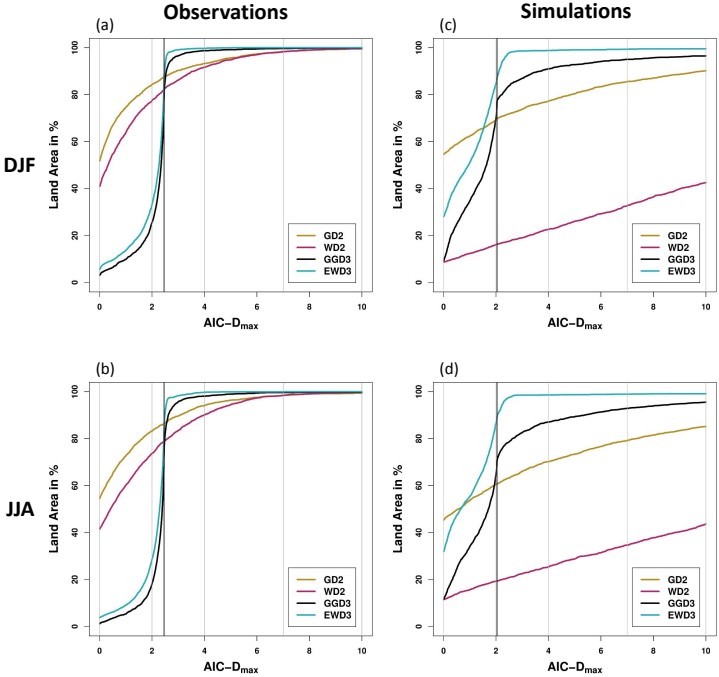

**Figure 5.** AIC-D frequencies: percentages of global land grid-points in which each distribution function yields AIC-D values that are smaller than or equal to a given AIC-D$_{max}$ value. The vertical black line indicates the different complexity penalties between 3- and 2-parameter PDFs. AIC-D frequencies are displayed for each candidate PDF for observations (**left**) and simulations (**right**) during DJF (**top**) and JJA (**bottom**).

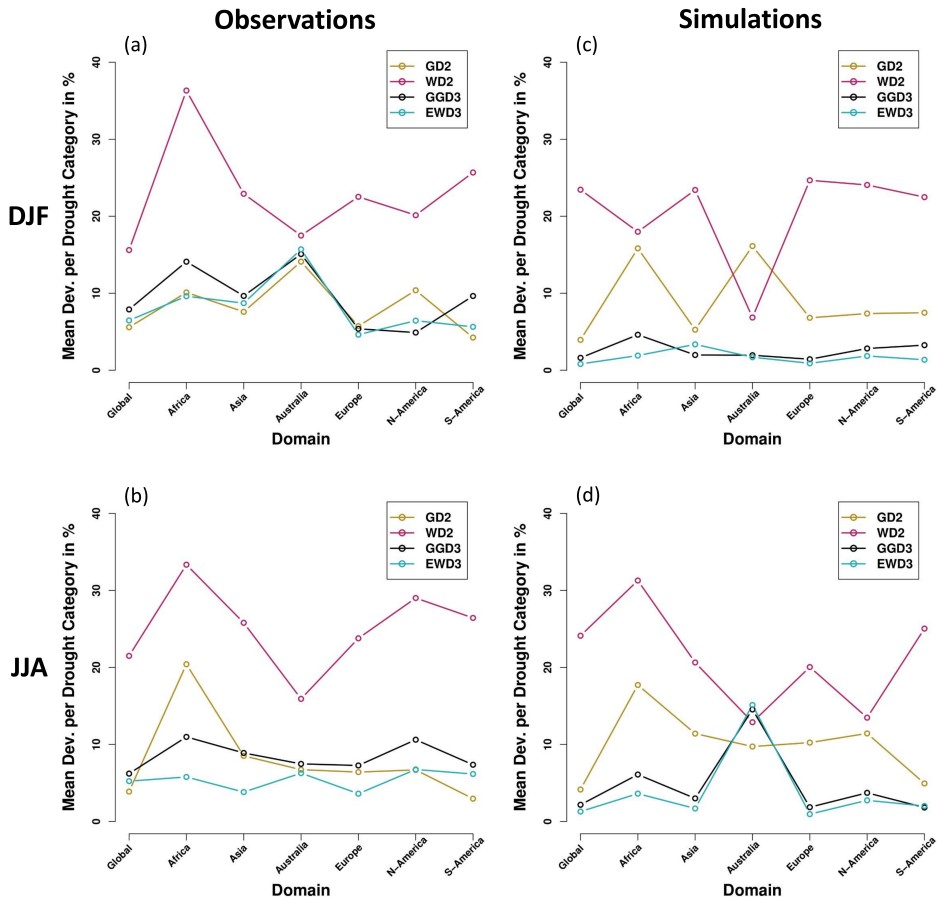

**Figure 6.** Mean deviations from $\mathcal{N}_{0,1}$ per SPI category for the entire global land area and each investigated region. Results are depicted for observations (**left**) and simulations (**right**) during DJF (**top**) and JJA (**bottom**).

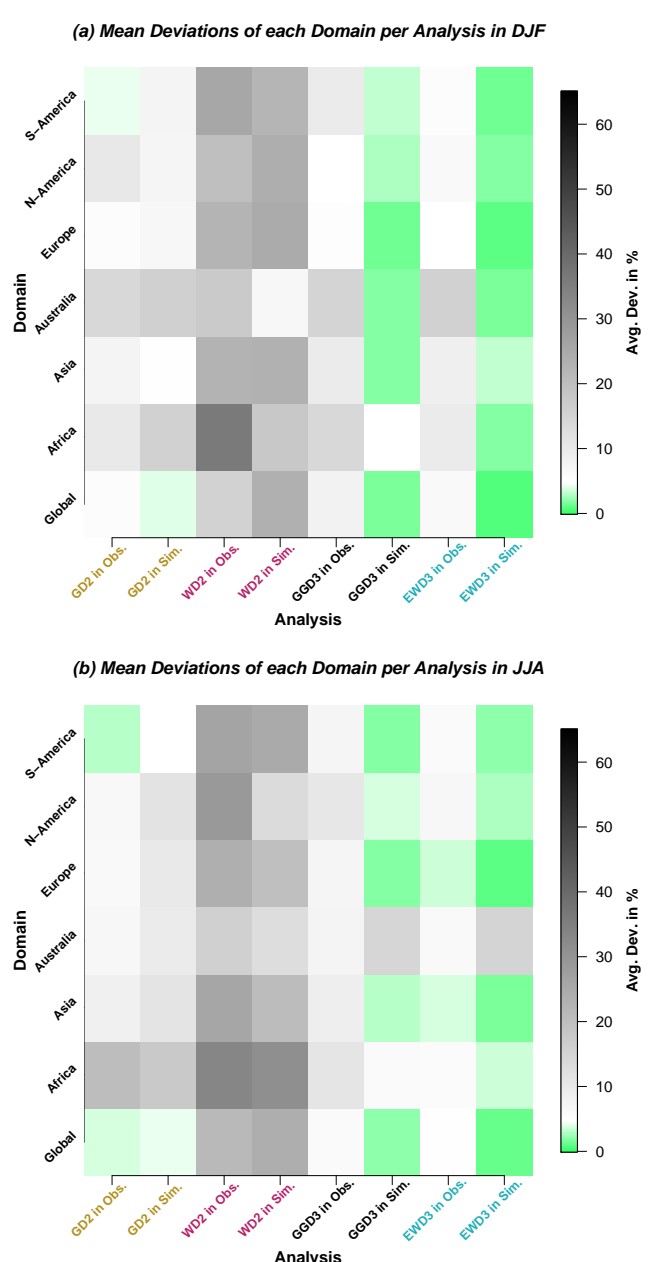

**Figure 7.** Mean deviations from $\mathcal{N}_{0,1}$ per SPI category during DJF (**a**) and JJA (**b**). Mean deviations are displayed for each investigated domain and each analyzed PDF for observations and simulations.

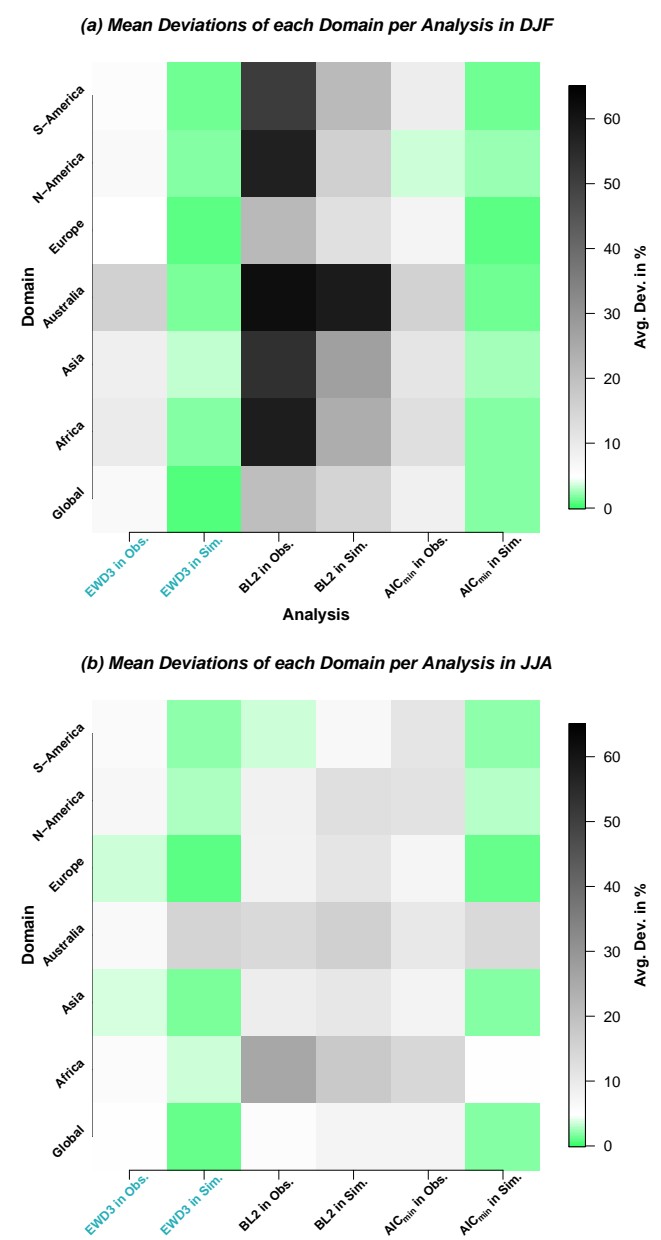

**Figure 8.** As in Fig. 7 but for the 3-parameter exponentiated Weibull distribution (EWD3) – the best performing candidate distribution function in this study –, a baseline which uses the 2-parameter gamma distribution (BL2), and a frequently proposed multi-PDF SPI calculation algorithm that uses in each grid-point and season that distribution function that yields in the respective grid-point and during the respective season the minimum AIC value ($AIC_{min}$-analysis which is denoted as $AIC_{min}$ in this figure). In contrast to GD2 in our previous analysis, BL2 employs a simpler optimization procedure of the same parameter estimation method (*maximum likelihood estimation*).