# Peer review of "A universal SPI candidate distribution function for observations and simulations"

_Hydrology and Earth System Sciences, 2019_

## Referee Comment (RC1) · Anonymous Referee #1 · 23 Jan 2020

General comments

The SPI (Standardized Precipitation Index) is a commonly and widely used index to detect droughts based on precipitation data. It can be applied to several aggregation periods of precipitation, e.g.1 month, 3 months, 6 months etc., tailored to the different drought impacts (meteorological drought, agricultural drought, hydrological drought, . . .). In doing so, a distribution function is fitted on the precipitation data and transformed to a standard distribution. This gives the possibility to detect and compare droughts over time and space. The curtail point is the reproduction of the standard distribution by the transformed original distribution. Here, the paper investigate the

suitability of four distribution functions with observed and forecasted precipitation data for the SPI. The goal of this paper is to propose one distribution function applicable to observed and forecasted precipitation totals globally for all useful aggregation periods. The paper is well and clear written and addresses the scientific question well.

Specific comments

- You wrote in lines 164 to 167 that you use three different procedures to estimate the parameters of the distribution function. Therefor I expect to get analyses of three procedures times four distributions equals to twelve analyses per observations and simulations. You showed only one per distribution. Which of the procedures did you used finally to fit the parameters of the distribution functions? This is also relevant as you wrote in section 3.1.3 that the procedure of estimation the distribution function parameters could have an impact on the usability of the derived parameters. - Do you exclude grids without converging parameter fits from the further analysis or to you use another procedure to estimate the parameters? Line 167/168 - Your sample sizes differ by a factor of ten between observations and forecasts (e.g. lines 198 or 277). In line 277, you wrote that the reliability of the parameters depends on the sample size and is therefore better for the modelled than for the observed data. Nevertheless, if you analyse the usability of distribution functions for the SPI, you should have parameter estimations with the same reliability. I propose to repeat the analysis with only one ensemble member and add that to the paper and add a short analysis on the impact of the available amount of data to the reliability of the SPI. - Lines 282 to 285: In this paragraph is no transition from absolute to relative AIC, which need to be improved. In addition, the index i is not well described. - Lines 224 to 226: Do you avoid parameters in the GGD3 to become GD2 or WB2? - Section 2.7: Your region are large enough to cover several precipitation regimes in one region. I propose to reduce the size of the regions and select regions with known good/bad model performance and different precipitation regimes. - Line 355: How do you calculate the "weighted sum"? Please add a description. - Line 574: You stated a phase transition of the SPI at 3 months

precipitation accumulation. However, I cannot see it in Figure 4. What do you mean with phase transition? - Section 4: Do you compare the same number of grid cells for observations and forecasts? In addition, do you compare the same grid cells? I assume different sets of selected grid cells for your analyses can have an impact on the results. - Lines 604/605: I think the investigations to the empirical cumulative distribution functions are very relevant for this topic and should be added to the paper or, at least, add a reference to the paper where you want to describe it. - Section 5: The base problem, from my point of view is, that the models are not able to reproduce the observed precipitation distribution function and procedures developed on observed data need to be adapted to be applied to model data (the GD2 performs well on the observed data). That is the base of your research and you should comment on this here or in the introduction. - Figure 6: Can you add the global average, as for Figure 4, as an additional domain to this figure?

Technical corrections

- Lines 379/380: It was not clear what was set in relation to what. Please reword this part. - Line 527: I think you to refer to Figure 8 and not to Figure 7. - Line 583: I think you want to refer to "GD2" instead of "GGD2" (typo). - Figure 4: Add to the caption that it is for global average.

---

## Referee Comment (RC2) · Gabriel Blain (Referee) · 9 Apr 2020

The manuscript "Global and regional performances of SPI candidate distribution functions in observations and simulations" proposes and new methodology to select candidates distributions for calculating the SPI; a widely used standardized drought index. The study is interesting and adds important information to the SPI literature because it evaluates the advantages and shortcomings of previous methodologies designed for the same purpose. It is also well written. So, it should be considered for publication.

I've just two minor suggestions: L.105 The Shapiro-Wilk [...] "is unreliable to evaluate SPI normality (Naresh Kumar et al., 2009)". This is a very important statement, which

now I tend to agree with. Please, provide further information regarding it.

The Bayesian information criterion (BIC) is similar to the AIC. However, the BIC uses a different penalty for the number of parameters [ln(n) k]. Can the authors verify if the BIC leads to similar results as those of the AIC.

---

## Author Comment (AC1) · 6 Jun 2020

**Response to Reviewer 2**

Patrick Pieper, André Düsterhus, and Johanna Baehr

June 6, 2020

We thank Gabriel Blain for the effort of reviewing our work. Your comments have been very helpful in improving our manuscript. Below we answer point-by-point to each of your comments and explain how the respective comment helped us to improve the manuscript. Your comments are printed in black and our responses are printed in blue. Line numbers in our response refer to the initially submitted manuscript.

One of your comments concerning the complexity punishment by our employed information criterion caused us to reconsider our storyline. This reconsideration does not alter our conclusions. Yet, it simplifies for us to conclude which helps readers to follow our conclusions.

**General comments**

The manuscript "Global and regional performances of SPI candidate distribution functions in observations and simulations" proposes and new methodology to select candidates distributions for calculating the SPI; a widely used standardized drought index. The study is interesting and adds important information to the SPI literature because it evaluates the advantages and shortcomings of previous methodologies designed for the same purpose. It is also well written. So, it should be considered for publication.

Thank you for these kind comments and your endorsement.

**Specific comments**

L.105 The Shapiro-Wilk [...] "is unreliable to evaluate SPI normality (Naresh Kumar et al., 2009)". This is a very important statement, which now I tend to agree with. Please, provide further information regarding it.

First, we are pleased that we were able to convince you. Second, we thank you for pointing out this lack of depth in our introduction.

Goodness-of-Fit (GoF) tests are ill-suited to assess the normality of SPI time-series, because of their spatial aggregation in combination with their binary convention. To fully understand this interplay we start with SPI's calculation

procedure: (i) fit a candidate PDF onto precipitation, then (ii) Z-transform the fitted probabilities to SPI values. Because the choice of an appropriate candidate PDF is the key decision in this process, the initial fit of the candidate PDF onto precipitation should be scrutinized. GoF tests, however, measure the normality of the resulting SPI values. In theory, this switch of focus in the analysis only complicates its structure but should not impact its outcome: if the candidate PDF's fit is appropriate, then its estimated probabilities are appropriate. Thus, their exact equiprobability transformations to the standard normal variable Z are also appropriate.

Anyhow, this complicated structure blurs the view on the measure of interest: the fit of the candidate PDF (onto precipitation). Therefore, the following caveat easily arises unnoticed and is, thus, not properly dealt with. After losing sight of the actual measure of interest (the fit of the candidate PDF), the focus lays on the normality of SPI time-series. The intuitive tool to assess normality leads to GoF tests. The drawback of GoF test is the biased discrimination between the tails and the center of the distribution. GoF tests equally evaluate each value that contributes to the distribution. Such an evaluation assigns more weight to the center and almost no weight to the tails of the distribution. Yet, appropriately fitting the tails of precipitation distributions should logically be of paramount importance to any sensible candidate PDF employed in SPI's calculation algorithm (see also our argument against weighting deviations from $\mathcal{N}_{0,1}$ by the theoretical occurrence probability of the respective class in lines 244 to 251). But the complicated structure blurs the view from this consideration. Instead, GoF tests conveniently present an allegedly easy solution.

As seen in Fig. 1, deviations from $\mathcal{N}_{0,1}$ are smallest in the center and largest in the tails of the distribution. Candidate PDFs typically fit precipitation better for the center than for the tails of the distribution: its center counts more samples which translates to more weight in the optimization (e.g. by the maximum likelihood estimation). This behavior deludes GoF tests in the analysis of SPI normality. That delusion obscures the tails of the distribution from GoF tests. Nevertheless this delusion, despite this obscurity surfacing skepticism about the proper depiction of the tails of the distribution can still aggregate over many grid-points. This aggregated skepticism can still lead to a robust analysis if evaluated relative to the similarly obscured performance of other candidate PDFs (as shown by metrics such as AIC-D, and BIC-D). Anyhow, additionally aggravating for GoF tests is their convention to be interpreted binarily. As a consequence of this convention, SPI literature typically aggregates results of GoF tests over domains by counting rejections. This typical aggregation prevents surfaced skepticism to fully aggregate over many grid-points. The interplay of both caveats, the blurred tails of the distribution and the prevention of remaining skepticism to fully aggregate, leads to the conclusion that GoF tests are ill-suited to assess SPI normality. I.e. it is (admittedly more obvious but) similarly inept to round normally distributed ($\mathcal{N}_{0.1\pm\epsilon,0.1}$) variables to their nearest integer before calculating their mean to estimate $\epsilon$.

This full explanation is too extensive for the scope of the introduction of

our publication. However, we do admit that only indicating problems with the binary nature of GoF tests and hinting at issues with their spatial aggregation might cut the story too short. To rectify this shortcoming, we split the paragraph (lines 106 -117). This allows us to elaborate on GoF tests (in)ability to evaluate SPI candidate distribution functions: "(...) which in turn is unreliable to evaluate SPI normality [Naresh Kumar et al., 2009].

The above-mentioned goodness-of-fit tests equally evaluate each value of SPI's distribution. Such an evaluation focuses on the center of the distribution because the center of any distribution contains per definition more samples than the tails. In contrast, SPI usually analyzes (and thus depends on a proper depiction of) the distribution's tails. Therefore, a blurred focus manifests in these goodness-of-fit tests. Moreover, the convention to binarily interpret the above-mentioned goodness-of-fit tests aggravates this blurred focus. Because of this convention, these goodness-of-fit tests are unable to produce any relative ranking of the performance of distribution functions for a specific location (and accumulation period). This inability prevents any reasonable aggregation of limitations that surface despite the blurred focus. Thus, they are ill-suited to discriminate the best performing PDF out of a set of PDFs [Blain et al., 2018]. For SPI distributions the question is not whether they are (or should be) normally distributed (for which goodness-of-fit tests are well suited to provide the answer). The crucial question is rather which PDF maximizes the normality of the resulting SPI distribution. Because of the ill-fitting focus and the ill-suited convention of these goodness-of-fit tests, they are inept to identify SPI's best-performing candidate distribution function out of a set of PDFs.

In agreement with this insight, those studies, that rigorously analyzed candidate distribution functions, or investigate an appropriate test methodology for evaluating SPI candidate PDFs, consequently advocate the use of relative assessments: (...)"

While elaborating on the methodology to test the normality of SPI time-series, we realized a missing differentiation between the analysis of AIC-D frequencies and the analysis of deviations from $\mathcal{N}_{0,1}$ in the initial submission. The fact that both analyses complement each other comes a bit too short. Thus, we also rectified this shortcoming through the following changes to the manuscript:

We substituted a sentence from the abstract in lines 6 to 7 by: "Our normality comparison bases on a complementary evaluation. Actual against theoretical occurrence probabilities of SPI categories evaluate the absolute performance of candidate distribution functions. In contrast, Akaike's information criterion evaluates candidate distribution functions relative to each other while analytically punishing complexity. SPI time-series (...)"

We added another paragraph at the end of section 2.5 in between lines 293 and 294 which reads as follows: "The analysis of deviations from $\mathcal{N}_{0,1}$ assesses performances of candidate PDFs in absolute terms irrespective of the candidate PDF's complexity. In contrast, the AIC-D analysis evaluates the performance of candidate PDFs relative to each other while analytically punishing complexity.

Consequently, the AIC-D analysis cannot evaluate whether the best-performing candidate distribution function also performs adequately in absolute terms. In opposition, deviations from $\mathcal{N}_{0,1}$ encounter difficulties when evaluating whether an increased complexity from one PDF to another justifies any given improvement. Both analyses together, however, augment each other complementary. This enables us to conclusively investigate: (i) which candidate PDF performs best while (ii) ensuring adequate absolute performance and while (iii) constraining the risk of over-fitting."

We substituted three sentences in a paragraph of section 3.1.1 (lines 401 to 405) by: "It is noteworthy, that investigating deviations from $\mathcal{N}_{0,1}$ over the entire globe contains the risk of encountering deviations that balance each other in different grid-points with unrelated climatic characteristics. Until dealing with this risk, our analysis of deviations from $\mathcal{N}_{0,1}$ only indicates that three candidate PDFs (GD2, GGD3, and EWD3) display an adequate absolute performance. On the one hand, we can reduce that risk by analyzing deviations from $\mathcal{N}_{0,1}$ only over specific regions. This analysis safeguards our investigation by ensuring (rather than just indicating) an adequate absolute performance around the globe and is performed later. On the other hand, we first completely eliminate this risk by examining AIC-D frequencies: aggregating AIC-D values over the entire globe evaluates the performance of PDFs in each grid-point and normalizes these evaluations by (rather than adding them over) the total number of grid-points of the entire globe. We investigate AIC-D frequencies first to evaluate whether GGD3 and/or EWD3 perform sufficiently better than GD2 to justify their increased complexities."

We added another paragraph at the end of section 3.1.1 (in between lines 475 and 476): "Among our candidate PDFs, EWD3 is obviously the best-suited PDF for SPI. Yet, we still need to confirm whether also EWD3's absolute performance is adequate. While the global analysis indicated EWD3's adequateness, the ultimate validation of this claim is incumbent upon the regional analysis."

We added another paragraph at the end of section 3.1.2 (in between lines 514 and 515): "The analysis of AIC-D frequencies proves that EWD3 is SPI's best distribution function among our candidate PDFs. Additionally, the regional investigation confirms the global analysis: the absolute performance of EWD3 is at minimum adequate in observations and ensemble simulations."

The Bayesian information criterion (BIC) is similar to the AIC. However, the BIC uses a different penalty for the number of parameters [ln(n) k]. Can the authors verify if the BIC leads to similar results as those of the AIC.

We thank you for this exciting idea. Whether we use AIC or BIC to punish candidate PDFs for their complexity does not change our conclusions. Most of our drawn conclusions from AIC-D frequencies bases on the behavior of candidate PDFs' coverages for AIC-$D_{max}$ values larger than 10 (right edge of Figure 5). These conclusions are then substantiated by candidate PDFs' coverages for AIC-$D_{max}$ values larger than 7. These coverages (for AIC-$D_{max}$/BIC-$D_{max}$

Table I. Complexity penalty of candidate PDFs assessed with AIC and BIC.

| Information Criterion | AIC | | BIC | | Difference BIC-AIC | |
|---|---|---|---|---|---|---|
| Realization | Obs. (N=31) | Sim. (N=310) | Obs. (N=31) | Sim. (N=310) | Obs. (N=31) | Sim. (N=310) |
| 2-param. PDFs | 4.43 | 4.04 | 6.87 | 11.47 | 2.44 | 7.43 |
| 3-param. PDFs | 6.89 | 6.08 | 10.3 | 17.21 | 3.41 | 11.13 |
| **Difference** 3-2 param. | **2.46** | **2.04** | **3.43** | **5.74** | **0.97** | **3.7** |

values $\leq 7$) are insensitive to the magnitude of changes caused by altered complexity penalties (Table I)

What impacts our analysis is not the absolute, parameter- and sample size-dependent punishment of candidate PDFs (values in the center of Table I). Instead, only the penalty difference between 2- and 3-parameter PDFs that base on the same sample size matters (evaluate observations and simulations isolated in the last row of Table I).

Similar, altering the information criterion (from AIC to BIC) impacts our analysis through the penalty difference between BIC and AIC (last column of Table I). Here, the difference between 2- and 3-parameter PDFs that base on the same sample size matters again (evaluate observations and simulations isolated in the bottom- and rightmost cell of Table I). I.e. the additional margin by which 3-parameter PDFs need to further outperform 2-parameter PDFs in order to still be considered as better by the new information criterion. This margin (bottom- and rightmost cell in Table I) increases in observations (simulations) by 0.97 (3.7) when using BIC instead of AIC.

The robustness of our conclusions stems from the robustness of the candidate's coverages for large AIC-$D_{max}$/BIC-$D_{max}$ values ($\geq 7$). In this AIC-$D_{max}$ regime, the candidate PDFs' coverages are sufficiently robust concerning changes caused by altered complexity penalties (Fig. I). Comparing 2- against 3-parameter in Fig. 5 with AIC-D or BIC-D does not substantially change the evaluation of large AIC-$D_{max}$/BIC-$D_{max}$ values ($\geq 7$). As a first-order approximation, we can compare in observations the coverages of 2-parameter PDFs at the AIC-$D_{max}$ value of 7 against the coverages of 3-parameter PDFs at the AIC-$D_{max}$ value 7.97 (we shift the line indicating the coverages of 3-parameter PDFs by 0.97 units to the right). Since the slope of that line is sufficiently flat, this shift does not impact the conclusions for large AIC-$D_{max}$ values ($\geq 7$).

In observations (simulations), coverages of 3-parameter PDFs are highly sensitive to the change of the information criterion at AIC-$D_{max}$/BIC-$D_{max}$ values smaller than approximately 4 (6) (compare in Fig I the top row against the bottom row). The first-order approximation outlined before (shifting the coverages of 3-parameter PDFs by 0.97 (3.7) units to the right in observations (simulations)), describes the changes caused by using BIC (instead of AIC) quite well. The shifted coverages of 3-parameter PDFs exhibit slope-dependent

[Figure]

Figure I. AIC-D (**top**) and BIC-D (**bottom**) frequencies: percentage of global grid-points during both seasons in which each PDF yields AIC-D/BIC-D values that are smaller than or equal to a given AIC-$D_{max}$/BIC-$D_{max}$ value. The vertical black line indicates the different complexity penalties between 3- and 2-parameter PDFs (see bottom row of Table I). AIC-D/BIC-D frequencies are displayed for each candidate PDF for observations (**left**) and simulations (**right**).

changes at all AIC-$D_{max}$/BIC-$D_{max}$ values. That causes 3-parameter PDFs to be best-suited (AIC-$D_{max}$/BIC-$D_{max}$ value of 0) in fewer grid-points. In each grid-point, a single PDF must still be best-suited. In a second-order approximation, the coverages of 2-parameter PDFs, thus, slightly adjust for small AIC-$D_{max}$/BIC-$D_{max}$ values to the changes of 3-parameter PDFs' coverages at the AIC-$D_{max}$/BIC-$D_{max}$ value of 0. Consequently, the coverages of 2-parameter PDFs are overall fairly insensitive to the change of the information criterion because they only adjust slightly. The coverages of 3-parameter PDFs are more sensitive to the changed information criterion because they universally exhibit a horizontal shift.

This shift, however, does not result in a universally uniform sensitivity. The sensitivity of the coverages of 3-parameter PDFs depends on their slope. Because their slope is in both realizations flat for AIC-$D_{max}$ values beyond 2.5, the coverages of 3-parameter PDFs are insensitive beyond AIC-$D_{max}$/BIC-

$D_{max}$ values of 2.5 plus 0.97 (3.7) in observations (simulations). Therefore, the coverages of the AIC-D/BIC-D category "no skill" (AIC-D/BIC-D > 10) and "sufficient" (AIC-D/BIC-D $\leq$ 7) are robust concerning a change of the information criterion from AIC to BIC in both realizations. In observations, the AIC-D/BIC-D category "well" (AIC-D/BIC-D $\leq$ 4) is also robust to the change of the information criterion (because $2.5 + 0.97 \leq 4$). Further, the slope of coverages of both 3-parameter PDFs is rather flat between AIC-$D_{max}$ values of 1 and 2, in observations. In observations, the AIC-D/BIC-D category "ideal" (AIC-D/BIC-D $\leq$ 2) is, therefore, also rather robust to the change of the information criterion. Ergo, all AIC-D/BIC-D categories are in observations sufficiently robust to the change of the information criterion. We identify sensitive performances to the change of the information criterion only in simulations for the AIC-D/BIC-D categories "ideal" and "well". This sensitivity does not affect the main argument against GD2 in simulations. GD2 displays a worthless (insufficient) performance in 12% (18%) of grid-points. Also for BIC-D frequencies, GD2 displays a worthless (insufficient) performance in more than 10% (14%) of grid-points in simulations. In contrast, EWD3 displays, irrespective of the employed information criterion, a worthless or insufficient performance only in 1% of grid-points – EWD3 reduces the count of grid-point characterized by this highly undesirable performance by over one magnitude.

We extensively draw our conclusion from erroneous performances of our candidate PDFs. Irrespective of the information criterion, erroneous performances are for EWD3 virtually non-existent, but manifest for GD2 in a non-negligible percentage of grid-points in both realizations. Thus, as also discussed in the initial submission (e.g. when introducing AIC-D in the results, and when elaborating on them in the discussion), the risk of underfitting by using 2-parameter PDFs seems larger than the risk of overfitting by using 3-parameter PDFs. Consequently, once the need for 3-parameter candidate PDFs is established, their remaining punishment relative to 2-parameter PDFs biases the analysis; particularly for small AIC-D values. Because of the complexity penalty in the information criterion, our 3-parameter candidate PDFs outperform our 2-parameter candidate PDF only for AIC-$D_{max}$ values beyond their increased complexity penalty (black vertical line in Fig I). We argue that maintaining the complexity penalty (beyond the proven inability of 2-parameter distributions) causes an artificial disadvantage for 3-parameter PDFs for small AIC-D values. Therefore, the complexity penalty biases and obscures our analysis for small AIC-$D_{max}$ values. We interpret the results from this BIC-D analysis as another confirmation of our line of argumentation. Anyhow, this discussion (and our interpretation of a confirmation of our line of argumentation) only underlines our conclusion that EWD3 is better suited than GD2. In contrast, we draw that conclusion from erroneous performances of GD2 that manifest irrespective of the employed information criterion.

The above-conducted analysis helped us to streamline our reasoning. In consequence, we slightly altered several lines of the manuscript to simplify our

line of argumentation. This helps us to convey, and readers to intuitively understand our conclusions. In this process, we conducted two different types of changes. Firstly, changes concerning the proper communication of AIC's punishment (including the above-mentioned bias). Secondly, changes that focus our analysis on GD2 and EWD3, instead of highlighting all four candidate PDFs almost equally prominent.

In the thorough analysis of AIC's and BIC's complexity penalties, we identified an intuitive way to visualize the penalty difference between 2- and 3-parameter PDFs. The black vertical line in Fig. I. Including this black line also in Fig. 5 enables us to elaborate more precise on the impact of that penalty difference. Therefore, we adapted Fig. 5 and discuss the adaptation in the text. This simplifies our line of argumentation.

We changed a paragraph in Section 3.1.1 (lines 458 to 470) to: "It seems worth elaborating on the insufficient (only average) confidence in EWD3 to perform ideally in observations (ensemble simulations) around the globe. The complexity penalty of AIC correctly punishes EWD3 stronger than GD2 because AIC evaluates whether EWD3's increased complexity (relative to GD2) is necessary. However, the results justify the necessity for this increased complexity – GD2 performs erroneously in 26% (6%), insufficiently in 18% (2%), and without any skill in 12% (1%) of the global land area in ensemble simulations (observations). The risk of underfitting by using 2-parameter PDFs seems larger than the risk of overfitting by using 3-parameter PDFs. Once the need for 3-parameter candidate PDFs is established, their remaining punishment relative to 2-parameter PDFs biases the analysis; particularly for the ideal AIC-D category. EWD3's increased complexity penalty relative to 2-parameter candidate PDFs depends on the sample size and amounts to 2.46 in observations and 2.04 in ensemble simulations (see black vertical lines in Fig. 5 (a)–(d)). The AIC-$D_{max}$ value beyond which EWD3 reaches coverages close to 100% approximately amounts to EWD3's increased penalty (see Fig. 5 (a)–(d)). Correcting EWD3's coverages for this bias would affect our evaluation of EWD3's performance only for the ideal AIC-D category. To illustrate this effect, we only consider AIC's estimated likelihood (without its penalty). Such a consideration corrects this complexity bias in EWD3's performance. While we analytically analyzed this consideration, a first-order approximation suffices for the scope of this publication. In that first-order approximation of this consideration, we simply shift the curve of EWD3 by 2.46 units leftwards in observations (Fig. 5 (a) and (b))) and by 2.04 units leftwards in ensemble simulations (Fig. 5 (c) and (d)). After this shift, EWD3 would also perform ideal with substantial confidence."

[revised manuscript text omitted]

---

## Author Comment (AC2) · 6 Jun 2020

**Response to Reviewer 1**

Patrick Pieper, André Düsterhus, and Johanna Baehr

June 6, 2020

We thank the reviewer for the effort of reviewing our work. His/Her comments have been very helpful in improving our manuscript. Below we answer point-by-point to each of the reviewer's comments and explain how the respective comment helped us to improve the manuscript. Reviewer's comments are printed in black and our responses are printed in blue. Line numbers in our response refer to the initially submitted manuscript.

One comment of the reviewer concerning the sample size in simulations caused us to perform a deeper sensitivity analysis on the ensemble size. In this process, a caveat to the drawn conclusions emerged. Therefore we include this sensitivity analysis to the results section and slightly adapted the drawn conclusions.

**General comments**

The SPI (Standardized Precipitation Index) is a commonly and widely used index to detect droughts based on precipitation data. It can be applied to several aggregation periods of precipitation, e.g.1 month, 3 months, 6 months etc., tailored to the different drought impacts (meteorological drought, agricultural drought, hydrological drought, . . .). In doing so, a distribution function is fitted on the precipitation data and transformed to a standard distribution. This gives the possibility to detect and compare droughts over time and space. The curtail point is the reproduction of the standard distribution by the transformed original distribution. Here, the paper investigate the suitability of four distribution functions with observed and forecasted precipitation data for the SPI. The goal of this paper is to propose one distribution function applicable to observed and forecasted precipitation totals globally for all useful aggregation periods. The paper is well and clear written and addresses the scientific question well.

Thank you for these kind comments and the effort of acquiring an in-depth understanding of our work.

**Specific comments**

You wrote in lines 164 to 167 that you use three different procedures to estimate the parameters of the distribution function. Therefor I expect to get analyses of three procedures times four distributions equals to twelve analyses per observations and simulations. You showed only one per distribution. Which of the procedures did you used finally to fit the parameters of the distribution functions? This is also relevant as you wrote in section 3.1.3 that the procedure of estimation the distribution function parameters could have an impact on the usability of the derived parameters.

Thank you for pointing out this unclear description of our methods. The three optimization methods referred to in lines 165 to 167 are used one after another. The goal is to find the most suitable parameters of the fit. To achieve this goal all available tools (all three optimization methods) are employed.

To avoid misunderstandings we performed the following changes to lines 161 to 170 in the manuscript: "(...) and dealt with later specifically. We estimate the parameters of our candidate PDFs in SPI's calculation algorithm with the maximum likelihood method [Nocedal and Wright, 1999] which is also the basis for the AIC computation.

Our parameter estimation method first identifies starting values for the $n$ parameters of the candidate PDFs by roughly scanning the $n$-dimensional phase-space spanned by these parameters. The starting values identified from that scan are optimized with the simulated annealing method (SANN) [Bélisle, 1992]. Subsequently, these by SANN optimized starting values are again further optimized by a limited-memory modification of the Broyden-Fletcher-Goldfarb-Shanno (also known as BFGS) quasi-Newton method [Byrd et al., 1995]. If the BFGS quasi-Newton method leads to a convergence of the parameters of our candidate PDF, we achieve our goal and end the optimization here. If the BFGS quasi-Newton method does not lead to a convergence of the parameters of our candidate PDF, then we circle back to the starting values optimized by SANN and optimize them again further but this time with the Nelder-Mead method [Nelder and Mead, 1965]. After identifying converging parameters, the probabilities of encountering the given precipitation totals are computed and transformed into cumulative probabilities $(G(x))$.

If neither the BFGS quasi-Newton nor the Nelder-Mead method leads to any convergence of the most suitable parameters of our candidate PDFs, then we omit these grid-points where convergence is not achieved. For the gamma, Weibull, and exponentiated Weibull distribution, non-converging parameters are rare exceptions and only occur in a few negligible grid-points. For the generalized gamma distribution, however, non-convergence appears to be a more common issue and occurs in observations as well as in simulations in roughly every fifth grid-point of the global land area. This shortcoming of the generalized gamma distribution needs to be kept in mind when concluding its adequacy in SPI's calculation algorithm.

Since PDFs that describe the frequency distribution of precipitation totals are required to be only defined for the positive real axis, (...)"

Do you exclude grids without converging parameter fits from the further analysis or to you use another procedure to estimate the parameters? Line 167/168

We excluded from our analysis those grid-points where we do not achieve any convergence. We also excluded grid-points where zero-precipitation events occurred more than one-third of the times in our time-period (see lines 188 to 189). Grid-points excluded through both of these reasons are mainly located in the Sahara. In the process of checking grid-points excluded from the analysis, we realized a misleading description in the manuscript concerning the excess of zero-precipitation events. While the simulated precipitation time-series of all ensemble members (n=310) exhibits in 3.68% of the global land grid-points too often (more than 103 times) zero-precipitation events, only a single grid-point (located in the Sahara) exhibits zero-precipitation events too often (more than 10 times) in observations (n=31). Barring one exception, all of the grid-points which exhibit zero-precipitation events too often in simulations are located in the Sahara and the Arabian Peninsula (9°N – 44°N; 16°E – 69°W). The only exception is one grid-point which is is located in the Nevada desert.

We clarified this asymmetry between observations and simulations in lines 189 to 191: "This limitation restricts the SPI calculation in simulations over the Sahara and the Arabian Peninsula for accumulation periods of 1- and 3-months, (...)"

Your sample sizes differ by a factor of ten between observations and forecasts (e.g. lines 198 or 277). In line 277, you wrote that the reliability of the parameters depends on the sample size and is therefore better for the modelled than for the observed data. Nevertheless, if you analyse the usability of distribution functions for the SPI, you should have parameter estimations with the same reliability. I propose to repeat the analysis with only one ensemble member and add that to the paper and add a short analysis on the impact of the available amount of data to the reliability of the SPI.

Thank you for this excellent idea. As a consequence of our focus on seasonal predictions (which heavily rely on the entire ensemble space), we did not recognize the possibility to potentially widen our conclusions through a sensitivity analysis of the sample size. As it turns out, differences between observations and simulations mostly evaporate while their main distinction results from the sample size. In contrast to observations, the sample size can easily be expanded or condensed in simulations through the employment of additional/fewer ensemble realizations.

EWD3 outperforms GD2 for a sample size of 31 years in simulations and observations (Table I). The better performance of EWD3 relative to GD2 is particularly important in those grid-points where GD2 does not perform well (AIC-D $\geq$ 4). EWD3 displays such an erroneous performance in virtually no

Table I. As in Table 3, but the evaluation of simulations bases on a single ensemble member. Observations are identical to Table 3.

| SPI Period | Realization | AIC-D category | GD2 | WD2 | GGD3 | EWD3 |
|---|---|---|---|---|---|---|
| 3-Months | Observations | Ideal (AIC-D $\leq 2$) | 84 | 76 | 22 | 31 |
| | | Well (AIC-D $\leq 4$) | 94 | 91 | 98 | 100 |
| | | Sufficient (AIC-D $\leq 7$) | 98 | 98 | 100 | 100 |
| | | No Skill (AIC-D $> 10$) | 1 | 0 | 0 | 0 |
| | Single Ensemble Member | Ideal (AIC-D $\leq 2$) | 83 | 76 | 19 | 28 |
| | | Well (AIC-D $\leq 4$) | 93 | 91 | 98 | 100 |
| | | Sufficient (AIC-D $\leq 7$) | 98 | 98 | 100 | 100 |
| | | No Skill (AIC-D $> 10$) | 1 | 0 | 0 | 0 |

grid-point. While these results still support our overall conclusions, it is evident that 2-parameter distribution functions can perform distinctly better in simulation than initially expected. The 2-parameter PDFs perform equally between observations and simulations. However, the 2-parameter PDFs also perform still worse than the 3-parameter PDFs. Yet, the insights gained from Table I also expose the question concerning the sensitivity of candidate PDFs' performances to the sample size.

Table II. As in Table 3, but with a focus on the sensitivity of the ensemble/sample size in simulations.

| SPI Period | Ensemble Size | AIC-D category | GD2 | WD2 | GGD3 | EWD3 |
|---|---|---|---|---|---|---|
| 3-Months | 2 | Ideal (AIC-D $\leq$ 2) | 78 | 56 | 43 | 57 |
| | | Well (AIC-D $\leq$ 4) | 87 | 74 | 96 | 99 |
| | | Sufficient (AIC-D $\leq$ 7) | 94 | 90 | 98 | 100 |
| | | No Skill (AIC-D $>$ 10) | 3 | 4 | 1 | 0 |
| | 3 | Ideal (AIC-D $\leq$ 2) | 77 | 45 | 53 | 69 |
| | | Well (AIC-D $\leq$ 4) | 86 | 61 | 96 | 99 |
| | | Sufficient (AIC-D $\leq$ 7) | 93 | 79 | 99 | 100 |
| | | No Skill (AIC-D $>$ 10) | 4 | 10 | 1 | 0 |
| | 4 | Ideal (AIC-D $\leq$ 2) | 75 | 38 | 59 | 74 |
| | | Well (AIC-D $\leq$ 4) | 84 | 50 | 95 | 99 |
| | | Sufficient (AIC-D $\leq$ 7) | 90 | 67 | 98 | 100 |
| | | No Skill (AIC-D $>$ 10) | 7 | 19 | 2 | 0 |
| | 5 | Ideal (AIC-D $\leq$ 2) | 74 | 31 | 63 | 79 |
| | | Well (AIC-D $\leq$ 4) | 82 | 42 | 94 | 99 |
| | | Sufficient (AIC-D $\leq$ 7) | 89 | 57 | 97 | 99 |
| | | No Skill (AIC-D $>$ 10) | 7 | 30 | 2 | 0 |
| | 6 | Ideal (AIC-D $\leq$ 2) | 73 | 27 | 64 | 80 |
| | | Well (AIC-D $\leq$ 4) | 81 | 36 | 93 | 99 |
| | | Sufficient (AIC-D $\leq$ 7) | 88 | 50 | 96 | 99 |
| | | No Skill (AIC-D $>$ 10) | 9 | 37 | 2 | 0 |
| | 7 | Ideal (AIC-D $\leq$ 2) | 70 | 25 | 66 | 81 |
| | | Well (AIC-D $\leq$ 4) | 78 | 33 | 92 | 98 |
| | | Sufficient (AIC-D $\leq$ 7) | 86 | 45 | 96 | 99 |
| | | No Skill (AIC-D $>$ 10) | 10 | 43 | 2 | 1 |
| | 8 | Ideal (AIC-D $\leq$ 2) | 69 | 21 | 67 | 83 |
| | | Well (AIC-D $\leq$ 4) | 77 | 29 | 91 | 98 |
| | | Sufficient (AIC-D $\leq$ 7) | 85 | 39 | 95 | 99 |
| | | No Skill (AIC-D $>$ 10) | 11 | 49 | 3 | 1 |
| | 9 | Ideal (AIC-D $\leq$ 2) | 66 | 20 | 67 | 85 |
| | | Well (AIC-D $\leq$ 4) | 76 | 27 | 90 | 99 |
| | | Sufficient (AIC-D $\leq$ 7) | 84 | 36 | 95 | 99 |
| | | No Skill (AIC-D $>$ 10) | 12 | 53 | 3 | 1 |

3-parameter PDFs benefit because of their increased complexity more than 2-parameter PDFs from an increased sample size which is realized by additional ensemble members (Table II). Consequently, reducing the ensemble size levels the playing field between 2- and 3-parameter PDFs. While a sample size of 31 years suffices EWD3 to outperform GD2, the margin by which EWD3 outperforms GD2 increases with a further increase in sample size.

Because of these insights, we rectified several statements in the manuscript which imply that 2-parameter PDFs are unable to sufficiently describe simulated precipitation. Instead, we emphasize that – despite the increased need of samples to fit 3 parameters – the 3-parameter distribution functions perform better than the 2-parameter PDFs among our candidate PDFs. This improved performance is already apparent for roughly 30 events and logically becomes more distinct with increasing sample size.

In view of these insights, we created subsection 3.1.4 (in between lines 562 and 563) in which we discuss Table I and Table II:

**"3.1.4 Sensitivity to Ensemble Size**

[revised manuscript text omitted]

Aside, we clarified the following statements of the manuscript:

We changed the wording from "simulations" to "ensemble simulations" in the following lines: 13, 362, 432, 450, 458, 495, 513, 569, 590, 665, 669, 693

We substituted the sentence in lines 462 to 464 by: "(...) However, the results justify the necessity for this increased complexity – GD2 performs erroneously in 26% (6%), insufficiently in 18% (2%), and without any skill in 12% (1%) of the global land area in ensemble simulations (observations). The risk of underfitting (...)"

We included the following paragraph in between lines 622 and 623: "Overall our 3-parameter candidate PDFs perform better than investigated 2-parameter

candidate PDFs. Despite requiring more data, a sample size of 31 years suffices our 3-parameter candidate PDFs to outperform our 2-parameter candidate PDFs in simulations and observations. Further, our 3-parameter candidate PDFs greatly benefit from an increase in the sample size in simulations. In simulations, such a sample size sensitivity analysis is feasible by exploiting different counts of ensemble members. Whether 3-parameter PDFs would benefit similarly from an increased sample size in observations is likely but ultimately remains speculative because trustworthy global observations of precipitation are temporally too constrained for such a sensitivity analysis."

We recalculated the counts in lines 656 to 659. They now read as follows: "Moreover, these thresholds show a robust statistical basis in terms of being equally represented over all 320 analyzed evaluations in this study (all entries of Table 3, Table 4. Table 5, and Table 6). Across all 80 analyses (all rows of Table 3, Table 4, Table 5, and Table 6), the four candidate PDFs perform insufficiently 132 times, while they perform with substantial (average) confidence 130 (58) times."

Lines 282 to 285: In this paragraph is no transition from absolute to relative AIC, which need to be improved. In addition, the index i is not well described.
Thank you for revealing this unclear description.

We changed lines 280 to 287 to: "(...) penalizes candidate PDFs based on their parameter-count. The best-performing distribution function attains the smallest AIC value because the first term is negative and the second one is positive.
Further, the absolute AIC value is often of little information – especially in contrast to relative differences between AIC values derived from different distribution functions. Thus, we use relative AIC differences (AIC-D) in our analysis. We calculate these AIC-D values for each PDF by computing the difference between its AIC value to the lowest AIC value of all four distribution functions. AIC-D values inform us about superiority in the optimal trade-off between bias and variance and are calculated as follows:

$$AIC\text{-}D_i = AIC_i - AIC_{min} \tag{1}$$

The index $i$ indicates different distribution functions. $AIC_{min}$ denotes the AIC value of the best-performing distribution function.
For our analysis, AIC-D values are well suited (...)"

Lines 224 to 226: Do you avoid parameters in the GGD3 to become GD2 or WB2?
We estimate the parameters of all PDFs independently by fitting the respective PDF to the precipitation data. Consequently, the two parameters that GGD3 share with GD2 (WD2) can differ. This is important because the third parameter of GGD3 (and EWD3) extends the phase-space spanned by the 2 parameters of GD2 (and WD2) into a third dimension. This third dimension provides opportunities for further optimizations – also for the first two parameters.

Thus, the new optimum for GGD3 in the three-dimensional phase-space does not need to be located along the normal above the optimal parameter-values of GD2 (or WD2) in the two-dimensional phase-space. The same is true in the other direction. The optimum location of parameters in the three-dimensional phase-space cannot simply be projected onto any two-dimensional phase-space. Instead, the location in the two-dimensional phase-space needs to be identified by properly optimizing the estimated fitting parameters independently.

To avoid misunderstandings, we clarified this point by inserting the following description at the end of the second paragraph of that section at line 211: "The optimization of this second shape parameter also requires the re-optimization of the first two parameters. The fitting procedure of 3-parameter PDFs needs therefore considerable more computational resources than the fitting procedure of 2-parameter distribution functions."

Section 2.7: Your region are large enough to cover several precipitation regimes in one region. I propose to reduce the size of the regions and select regions with known good/bad model performance and different precipitation regimes.

As of yet, the analysis is condensed enough to display the regional results in Figures 6-8 in single plots. Such a visualization helps to convey the results of our analysis. Until now, we presumed our results to be sufficiently robust so that the exact borders of our regions would neither distinctly alter our results nor our conclusions. Aside, the analyzed regions need to encompass several grid-points as explained in lines 322 to 327. Adhering to the *law of large numbers* is crucial for the statistical analysis performed for reach region. That being said, one can still argue for smaller regions. However, such a dispute is subjective as described in lines 330 to 324. Resolving this dispute would lead to an entirely new analysis which is beyond the scope of this investigation.

Irrespective of resolving this dispute in general, your proposal also triggered our curiosity concerning our presumption about the spatial robustness of our results and conclusions. Therefore, we tested the analysis for a region with exceptionally good performance of MPI-ESM-LR in predicting precipitation and SPI: the *North Region* of Brazil (0°– 8°S; 40°W – 60°W). As a side note, examples of poor model performance are already included in the results (e.g. the entire European continent). For the *North Region* of Brazil, we repeated Figure 4 and Table 3 of our analysis and display these results in Figure I and Table III.

Table III. As in Table 3, but solely for the *North Region* of Brazil (0°– 8°S; 40°W – 60°W).

| SPI Period | Realization | AIC-D category | GD2 | WD2 | GGD3 | EWD3 |
|------------|-------------|----------------|-----|-----|------|------|
| 3-Months | Observations | Ideal (AIC-D ≤ 2) | 69 | 76 | 12 | 35 |
| | | Well (AIC-D ≤ 4) | 84 | 89 | 92 | 100 |
| | | Sufficient (AIC-D ≤ 7) | 100 | 97 | 100 | 100 |
| | | No Skill (AIC-D > 10) | 0 | 0 | 0 | 0 |
| | Simulations | Ideal (AIC-D ≤ 2) | 13 | 50 | 70 | 93 |
| | | Well (AIC-D ≤ 4) | 13 | 53 | 84 | 100 |
| | | Sufficient (AIC-D ≤ 7) | 16 | 77 | 87 | 100 |
| | | No Skill (AIC-D > 10) | 78 | 21 | 8 | 0 |

[Figure]

Figure I. As in Figure 4, but solely for the *North Region* of Brazil (0°– 8°S; 40°W – 60°W).

On the one hand, these results further corroborate our conclusions. EWD3 is distinctly better suited than the other candidate PDFs to describe precipitation; also when analyzed over such a small region (see Table III). On the other hand, the results also exemplify the importance of adhering to the *law of large numbers* in our analysis and its sensibility in terms of the extend of analyzed regions; specifically when evaluating deviations from $\mathcal{N}_{0,1}$ (see Figure I).

Line 355: How do you calculate the "weighted sum"? Please add a description.

Thank you for pointing out this lack of clarity.

We changed the sentence to: "Therefore, the weighted sum (weighted by the theoretical occurrence probability of the respective SPI class (Table 2)) over the absolute values of deviations from $\mathcal{N}_{0,1}$ along all SPI categories is lowest for GD2 in both analyzed seasons (see legend in Fig. 4, (a)–(d))."

We also added another description in line 377: "Therefore, we weigh each class' deviation from $\mathcal{N}_{0,1}$ by the theoretical occurrence probability (see Table 2) of the respective class and analyze weighted deviations from $\mathcal{N}_{0,1}$."

Line 574: You stated a phase transition of the SPI at 3 months precipitation accumulation. However, I cannot see it in Figure 4. What do you mean with phase transition?

Thank you for calling the misleading phrasing to our attention. In Table 4, WD2 performs better than GD2 in observation for an accumulation period of 1-month. For accumulation periods of 6-months and longer GD2 performs better than WD2 in observations.

We see how referring to this behavior as phase transition might be misleading and changed the paragraph to: "In agreement with prior studies [Stagge et al., 2015, Sienz et al., 2012], we also identify the apparent performance shift between short (less than 3-months) and long (more than 3-months) accumulation periods for the 2-parameter candidate PDFs. While WD2 performs well for short accumulation periods (only in observations though), GD2 performs better than WD2 for longer accumulation periods. Nevertheless, neither 3-parameter candidate PDF displays such a shift in its performance. Both 3-parameter PDFs perform for accumulation periods shorter and longer than 3-months similarly well."

We also changed the sentence from line 600 to 602 in which we also used the wording *phase transition*. The reworded sentence reads as follows: "The emergence of this proposal stems from a focus on 2-parameter PDFs that exhibit a shift in their performance which depends on the scrutinized accumulation period."

Section 4: Do you compare the same number of grid cells for observations and forecasts? In addition, do you compare the same grid cells? I assume

Table IV. Percent of covered global land grid-points for each PDF in each realization and for each investigated season. Main differences between observations and simulations result from the Sahara and the Arabian Peninsula not being covered in simulations.

|  |  | GD2 | WD2 | GGD3 | EWD3 |
|---|---|---|---|---|---|
| | **Simulations** | 96.27 | 96.27 | 82.69 | 95.23 |
| DJF | **Observations** | 100.00 | 100.00 | 82.33 | 97.16 |
| | **Ratio: Sim./Obs.** | 0.9627 | 0.9627 | 1.0043 | 0.9801 |
| | **Simulations** | 96.33 | 96.33 | 77.9 | 95.55 |
| JJA | **Observations** | 99.97 | 99.97 | 84.79 | 96.87 |
| | **Ratio: Sim./Obs.** | 0.9636 | 0.9636 | 0.9187 | 0.9864 |

different sets of selected grid cells for your analyses can have an impact on the results.

Thank you for this well-founded remark. Because of this comment and an earlier comment of yours, we double-checked the omitted grid-points again. We omit grid-points because of excessive zero-precipitation events and as a result of not achieved convergences. Consequently, the analyzed grid-points differ. They differ between simulations and observations because both realizations exhibit a different count of grid-points which exhibited too many (more than one-third) zero-precipitation events. Additionally, the analyzed grid-points also differ across the analyzed PDFs because the count of grid-points in which convergence is not achieved varies PDF-dependent. It is noteworthy, that (for GD2, WD2, and EWD3) the variations in analyzed grid-points are dominated by excessive zero-precipitation events; rather than being caused by non-converging parameters. Averaged over both seasons, 3.68% (0%) of land grid-points are PDF-independently excluded through an excessive count of zero-precipitation events in simulations (observations). In contrast, the total percentage of omitted grid-points per PDF (as a result of non-convergence and excessive zero-precipitation events) are displayed in Table IV.

We excluded non-converging grid-points only for the specific PDF, the specific season, and only in the specific realization (observation or simulation). This results in slightly different coverages for each PDF and each realization (see Table IV). Admittedly, GGD3's coverage can be described as inferior compared to the other candidate PDFs. However, this inferior performance does not impact our conclusions, but rather affirms the conclusion that EWD3 is better suited than GGD3. Additionally, the similar coverages of the other three candidate PDFs support the claim of a leveled playing field in our analysis. Thus, repeating the analysis for those grid-points where the fits of GD2, WD2, and EWD3 mutually converge is highly unlikely to change the result. Moreover, limiting the analyzed grid-points to those grid-points in which GGD3's calculation algorithm finds converging parameters would artificially reduce the reliability of the comparison between GD2, WD2, and EWD3. This impact would be similarly undesirable.

Yet, we do agree that different sets of grid-points can principally impact our analysis. Therefore, we analyzed Table 3 again to ascertain our assumption of a negligible impact on our analysis:

Table V. As in Table 3, but only for those grid-points which are mutually covered in simulations and observations by each PDF. Note: Grid-point coverage still differs between DJF and JJA. Depicted is the mean over both seasons.

| SPI Period | Realization | AIC-D category | GD2 | WD2 | GGD3 | EWD3 |
|---|---|---|---|---|---|---|
| 3-Months | Observations | Ideal (AIC-D ≤ 2) | 84 | 74 | 19 | 30 |
| | | Well (AIC-D ≤ 4) | 94 | 90 | 98 | 100 |
| | | Sufficient (AIC-D ≤ 7) | 98 | 98 | 100 | 100 |
| | | No Skill (AIC-D > 10) | 0 | 0 | 0 | 0 |
| | Simulations | Ideal (AIC-D ≤ 2) | 64 | 18 | 68 | 86 |
| | | Well (AIC-D ≤ 4) | 73 | 24 | 89 | 99 |
| | | Sufficient (AIC-D ≤ 7) | 82 | 34 | 94 | 99 |
| | | No Skill (AIC-D > 10) | 12 | 56 | 4 | 1 |

Table VI. As in Table 3, but only for those grid-points which are mutually covered in simulations and observations by GD2, WD2, and EWD3. Note: Grid-point coverage still differs between DJF and JJA. Depicted is the mean over both seasons. Remark: Grid-points analyzed for GGD3 are the ones from Table 3 minus those grid-points which are not mutually covered by GD2, WD2, and EWD3.

| SPI Period | Realization | AIC-D category | GD2 | WD2 | GGD3 | EWD3 |
|---|---|---|---|---|---|---|
| 3-Months | Observations | Ideal (AIC-D ≤ 2) | 84 | 75 | 20 | 30 |
| | | Well (AIC-D ≤ 4) | 94 | 91 | 98 | 100 |
| | | Sufficient (AIC-D ≤ 7) | 98 | 98 | 100 | 100 |
| | | No Skill (AIC-D > 10) | 0 | 0 | 0 | 0 |
| | Simulations | Ideal (AIC-D ≤ 2) | 65 | 18 | 68 | 86 |
| | | Well (AIC-D ≤ 4) | 74 | 24 | 89 | 99 |
| | | Sufficient (AIC-D ≤ 7) | 82 | 34 | 94 | 99 |
| | | No Skill (AIC-D > 10) | 12 | 57 | 4 | 1 |

Averaged over all 32 entries, Table VI (Table V) differs on average by just 0.16 (0.34) percentage points from Table 3. The largest difference emerges in observations for GGD3 in the ideal category which deviates in Table VI (Table V) by 2 (3) percentage points from Table 3. In conclusion, we consider our assumption of a negligible impact on our analysis ascertained.

Lines 604/605: I think the investigations to the empirical cumulative distribution functions are very relevant for this topic and should be added to the paper or, at least, add a reference to the paper where you want to describe it.
We tried the empirical cumulative density function (ECDF) but quickly realized its shortcoming: Its discrete nature is too coarse for the task at hand which results in a massive dependence of possible SPI-values on the sample size. As explained in lines 323 to 328, the crucial performance requirement demands that deviations from $\mathcal{N}_{0,1}$ spatially balance each other sufficiently quickly. For SPI time-series derived with an ECDF, however, these deviations will never balance each other but aggregate with each additional grid-point. In the example from line 325, SPI time-series derived with an ECDF would not lead in a single grid-point to an extremely dry/wet event and would lead in each grid-point to exactly one severely dry/wet event during a 31-year time-series. Thus, for each grid-point over which we aggregate, we would add 0.7 missing extreme events and 0.4 missing severe events on both tails of the distribution.

To prevent any confusion, we adjusted the ending of the sentence in line 607 and included another explanation: "(...) We checked this approach which proved to be too coarse because of its discretized nature (not shown). As a result of its discretized nature, the analyzed sample size prescribes the magnitude of deviations from $\mathcal{N}_{0,1}$. Consequently, these deviations are spatially invariant and aggregate with each additional grid-point. Thus, deviations from $\mathcal{N}_{0,1}$ will not spatially balance each other."

Section 5: The base problem, from my point of view is, that the models are not able to reproduce the observed precipitation distribution function and procedures developed on observed data need to be adapted to be applied to model data (the GD2 performs well on the observed data). That is the base of your research and you should comment on this here or in the introduction.
Thank you for pin-pointing this motivation. This is exactly the motivation we had in mind which triggered us to conduct this analysis. We thought that we sufficiently pointed that out. However, after re-reading the respective paragraphs, we also realized that it comes a bit short. Therefore, we adjusted the Introduction and Section 5 and address this motivation in separate, stand-alone paragraphs:

To adjust the Introduction, we split the paragraph from lines 118 to 134. The changes read as follows: "SPI calculation procedures were developed for observed precipitation data. Since models do not exactly reproduce the observed precipitation distribution, these procedures need to be tested and eventually

adapted before being applied to modeled data. Here, we aspire to identify an SPI calculation algorithm that coherently describes modeled and observed precipitation (i.e. describes both modeled and observed precipitation distributions individually and concurrently). While testing SPI's calculation algorithm on modeled precipitation data is usually neglected, such a test demands nowadays a similarly prominent role as the one for observations because of the increasing importance of drought predictions and their evaluation. Despite this importance, the adequacy of different candidate distribution functions has to the authors' best knowledge never been tested in the output of a seasonal prediction system – although seasonal predictions constitute our most powerful tool to predict individual droughts. To close that gap, this study evaluates the performance of candidate distribution functions in an output of 10 ensemble members of initialized seasonal hindcast simulations.

In this study, we test the adequacy of the gamma, Weibull, generalized gamma, and exponentiated Weibull distribution in SPI's calculation algorithm. The evaluation of their performance depends on the normality of the resulting SPI time-series. In this evaluation, we focus on an SPI accumulation period of 3-months ($\text{SPI}_{3M}$) during winter (DJF) and summer (JJA) and test the drawn conclusions for other common accumulation periods (1-, 6-, 9-, and 12-months). Our analysis conducts two complementary evaluations of their normality: (i) evaluating their normality in absolute terms by comparing actual occurrence probabilities of SPI categories (as defined by WMO's *SPI User Guide* [Svoboda et al., 2012]) against well-known theoretically expected occurrence probabilities from the standard normal distribution ($\mathcal{N}_{0,1}$), (ii) evaluating their normality relative to each other with Akaike's information criterion (AIC) which analytically assesses of the *optimal trade-off* between information gain against the complexity of the PDF to adhere to the risk of overfitting. During this analysis, we investigate observations and simulations. Observed and simulated precipitation is obtained from the monthly precipitation data-set of the Global Precipitation Climatology Project (GPCP) and the above mentioned initialized seasonal hindcast simulations, respectively. We conduct our analysis for the period 1982 to 2013 with a global focus which also highlights regional disparities on every inhabited continent (Africa, Asia, Australia, Europe, North America, and South America)."

To adjust Section 5, we inserted in between Lines 672 and 673 (at the start of the section) the following paragraph: "Current SPI calculation algorithms are tailored to describe observed precipitation distributions. Consequently, current SPI calculation algorithms are ineptly suited to describe precipitation distributions obtained from ensemble simulations. Also in observations, erroneous performances are apparent and well-known, but less conspicuous than in ensemble simulations. We propose a solution that rectifies these issues and improves the description of modeled and observed precipitation distributions individually as well as concurrently. The performance of 2-parameter candidate distribution functions is inadequate for this task. By increasing the parameter count of the candidate distribution function (and thereby also its complexity) a distinctly

better description of precipitation distributions can be achieved. In simulations and observation, the here identified best-performing candidate distribution function – the exponentiated Weibull distribution (EWD3) – performs proficiently for every common accumulation period (1-, 3-, 6-, 9-, and 12-months) virtually everywhere around the globe. Additionally, EWD3 excels when analyzing ensemble simulations. Its increased complexity (relative to GD2) leads to an outstanding performance of EWD3 when an available ensemble multiplies the sample size."

Figure 6: Can you add the global average, as for Figure 4, as an additional domain to this figure?

We agree that the global average belongs in this Figure. To avoid any confusion, we decided to prominently label the global average in the caption of the figure.

The caption now reads as follows: "Mean deviations from $\mathcal{N}_{0,1}$ per SPI category for the entire global land area and each investigated region. Results are depicted for observations (**left**) and simulations (**right**) during DJF (**top**) and JJA (**bottom**)."

**Technical corrections**

Lines 379/380: It was not clear what was set in relation to what. Please reword this part.

Corrected.

Reworded to: "Relative to observations, GD2's weighted deviations increase in simulations by more than 120% in JJA, while WD2's increase by more than 25% in JJA and 80% in DJF."

Line 527: I think you to refer to Figure 8 and not to Figure 7.

We do mean Figure 7.

To clarify this misunderstanding, we reworded the sentence to: "The comparison between the performance of our baseline against GD2's performance (compare Fig. 8 against Fig. 7) thus also indicates the impact of the meticulousness applied to the optimization of the same parameter estimation method."

Line 583: I think you want to refer to "GD2" instead of "GGD2" (typo).

We want to refer to GGD3. We corrected that typo and changed "GGD2" to "GGD3".

Figure 4: Add to the caption that it is for global average.

Added.

---

## Author Response (AR2)

**Response to the Editor**

Patrick Pieper, André Düsterhus, and Johanna Baehr

July 6, 2020

Dear Marie-Claire ten Veldhuis,

We thank you for the effort of overseeing and actively participating in the review process. Your engagement has considerably contributed to the improvement of our manuscript. Below we answer point-by-point to each of your comments and explain how the respective comment helped us to improve the manuscript. Your comments are printed in black and our responses are printed in blue. Line numbers in our response refer to the post-referee version of the manuscript.

**General comments**

Comments to the Author:

Thanks for making the improvements to the manuscript, the referee comments have been addressed adequately.

I have a few remaining minor remarks related to the present version:

Thank you for your approval of our replies to the referee comments. Furthermore, we greatly appreciate your effort to actively participate in the review process. Your comments demonstrate a keen grasp of our work. Consequently, your well-founded suggestions particularly helped us to improve the clarity of our message. We are deeply grateful for this invaluable contribution and like to explicitly express this gratitude to you.

**Specific comments**

1. Title: the current title gives the impression the paper is an evaluation study while in fact you provide a solution for selecting the appropriate distribution for SPI calculation. Consider revising to make this innovative aspect of the paper more apparent.

We completely agree with your comprehension of our work. While evaluating different SPI candidate distribution functions, we identify that one of them – the exponentiated Weibull distribution (EWD3) – displays an exceptionally promising performance. Thus, EWD3 shows great potential to solve the long-standing dispute about SPI's most appropriate candidate PDF.

For the title to reflect this aspect, we changed the wording of the title from "Global and regional performances of SPI candidate distribution functions in observations and simulations" to:

**A universal SPI candidate distribution function for observations and simulations**

Consequntly, we also changed the wording of the running title from "SPI Candidate Distribution Functions" to:

**Universal SPI candidate distribution function**

2. Abstract: the quality and content of the abstract no longer matches that of the paper, after the improvements that have been made. The phrasing in the main manuscript is much better than in the abstract. Please check and revise the abstract text.

Two specific points to consider for the revision of the abstract:

- addition of quantitative details on how much better the exp. Weibull distribution performs compared to others

- explicitly mention that distributions were tested for both observed and modelled precipitation.

We thank you for calling this asymmetry to our attention. We agree that particularly the last paragraph of the abstract incomprehensively expresses the full scope of the study's main conclusion.

We changed the last paragraph of the abstract (lines 14-20) as follows to rectify this shortcoming:

"Our results suggest that calculating SPI with the commonly used gamma distribution leads to deficiencies in the evaluation of ensemble simulations. Replacing it with the exponentiated Weibull distribution reduces the area of those regions, where the index does not have any skill for precipitation obtained from ensemble simulations by more than one magnitude. The exponentiated Weibull distribution maximizes also the normality of SPI obtained from observational data and a single ensemble simulation. We demonstrate that calculating SPI with the exponentiated Weibull distribution delivers better results for each continent and every investigated accumulation period, irrespective of the heritage of the precipitation data. Therefore, we advocate the employment of the exponentiated Weibull distribution as the basis for SPI."

We also changed the last sentence of the abstract's second paragraph (lines 11-13). This sentence emphasizes now more accurately the study's focus: different performances of SPI candidate PDFs in observations and simulations. The sentence reads now as follows: "(...) While focusing on regional performance disparities between observations and simulations that manifest in an accumulation period of 3-months, we additionally test the drawn conclusions for other common accumulation periods (1-, 6-, 9-, and 12-months)."

3. It looks there is no discussion of the performance of distributions across SPI categories (presented in Figure 4). If i understand correctly, these reflect

the center and tails of the distribution (not explained in the Figure caption, unfortunately)? This is an important aspect that merits separate evaluation (i.e; dedicate a paragraph to it in the manuscript).

We thank you for pointing out this neglected aspect. The aspect is indeed important and adds to our story.

Consequently, we split the first paragraph of section 4 to discuss candidate PDFs' representations of the center and the tails of SPI's distribution. We inserted this discussion at line 684:

"(...) SPI's calculation algorithm needs to capture sufficiently well both frequency distributions mutually: those of observed and modeled precipitation totals.

The outlined problem is additionally aggravated by the fact that it cannot be circumnavigated. Our results demonstrate that any inept description of precipitation by SPI's candidate distribution function manifests most severely in the tails of SPI's distribution. Since SPI is usually employed to analyze the left-hand tail of its distribution (droughts), biased descriptions of this tail are highly undesirable. To establish the robustness of this valuable tool and to fully capitalize its advantages, SPI's problem of requiring a single, universally applicable candidate PDF needs to be solved. In this study, we show that the 3-parameter exponentiated Weibull distribution (EWD3) is very promising in solving this problem virtually everywhere around the globe in both realizations (observations and simulations) for all common accumulation periods (1-, 3-, 6-, 9-, and 12-months).

Other studies have dismissed the possibility of such a solution (...)"

We also added a short elaboration to the caption of Fig 3. This caption now reads as follows:

"Deviations from $\mathcal{N}_{0,1}$ over the entire globe for observed (**left**) and modeled (**right**) SPI time-series. SPI time-series are derived by using the simple 2-parameter gamma distribution (GD2, **top row**), the simple 2-parameter Weibull distribution (WD2, **second row**), the 3-parameter generalized gamma distribution (GGD3, **third row**), and the 3-parameter exponentiated Weibull distribution (EWD3, **bottom row**). The legends depict weighted (by their respective theoretical occurrence probability) sums (WS) of deviations from $\mathcal{N}_{0,1}$ over all SPI categories. Irrespective of the candidate PDF, deviations from $\mathcal{N}_{0,1}$ are smallest for the center of SPI's distribution (N0) and largest for its tails."

4. the phrasing "bases on" is incorrect and should be replaced by "is based on", throughout the manuscript.

Thank you for informing us about our, in this instance, misguided thrive to avoid passive language.

We corrected the wording in lines: 6, 101, 152, and 630. Aside, we also corrected the wording in the last sentence of the captions of Table 4, and Table 5.

117      The new revised version of the manuscript will be subject to review only by
118  the editor.
119      We are looking forward to your decision.

[revised manuscript text omitted]